# Seasonal Evolution of the Sea Ice Floe Size Distribution in the Beaufort Sea from Two Decades of MODIS Data

Ellen M. Buckley[1], Leela Cañuelas[1], Mary-Louise Timmermans[2], and Monica M. Wilhelmus[1]

[1]Center for Fluid Mechanics, School of Engineering, Brown University, Providence, RI, USA
[2]Department of Earth and Planetary Sciences, Yale University, New Haven, CT, USA

**Correspondence:** mmwilhelmus@brown.edu

**Abstract.** The Arctic sea ice cover seasonally evolves from large plates separated by long, linear leads in the winter to a mosaic of smaller sea ice floes in the summer. The interplay between physical and thermodynamic mechanisms during this process ultimately sets the observed sea ice floe size distribution (FSD), an important metric for characterizing the sea ice cover and assessing model performance. Historically, FSDs have been studied at fixed locations over short periods, leaving a gap in our understanding of the spatial and temporal evolution of the FSD at large scales. Here, we present an automated framework for image segmentation, allowing the identification and labeling of individual ice floes in Moderate Resolution Imaging Spectro-radiometer (MODIS) data. Using this algorithm, we automatically process and segment 4,861 images, identifying more than 9.4 million floes over 23 years. The extracted characteristics of the floes–including area, perimeter, and orientation–evolve throughout the spring and summer in the Beaufort Sea. We find seasonal patterns of decreasing mean floe area, increasing FSD power law slope, and more variability in the floe orientation as the summer progresses.

## 1 Introduction

The Arctic sea ice cover controls heat and moisture flux between the atmosphere and the ocean. It has an annual cycle characterized by the growth and melt of ice in which large, heterogeneous snow-covered winter ice floes fragment into an ensemble of smaller floes in the summer. The break-up of the ice cover and resulting floe size distribution (FSD) is set by complex feedback loops involving physical and thermodynamic processes. For instance, given the higher perimeter-to-area ratios of small floes compared to that of larger floes, small floes experience proportionally more lateral melting. This effect is observed to be especially pronounced for floes smaller than 50 m in diameter (Steele, 1992; Horvat et al., 2016). Lateral melting shrinks the floes, further raising their perimeter-to-area ratios, thus leading to a positive feedback cycle. Also, lateral melt creates density gradients that contribute to the non-homogeneous stratification of the upper ocean mixed-layer, enhancing mixing and eddy formation (Horvat et al., 2016). Long waves from summer Arctic storms fracture the ice pack (Asplin et al., 2012), an effect amplified by the retreating sea ice edge and lengthened open water fetch enhancing wave energy. The FSD also yields information about how a sea ice field will respond to oceanic and atmospheric forcing. Since the drag coefficient between an ice floe and the ocean depends on the ice floe size, the FSD is related to ocean-atmosphere energy and momentum transfer (Birnbaum and Lüpkes, 2002).

Following the original conceptualization of the FSD in Rothrock and Thorndike (1984), numerous studies have documented FSDs in various regions throughout the Arctic and the Antarctic (see Stern et al. (2018b) for a comprehensive list of FSD studies). We focus on the Beaufort Sea, where FSDs have previously been determined from radar imagery (Holt and Martin, 2001; Hwang et al., 2017), high-resolution optical satellite imagery (Wang et al., 2016; Denton and Timmermans, 2022), and aerial photography (Rothrock and Thorndike, 1984; Perovich and Jones, 2014). While these studies have advanced our understanding of the FSD seasonal evolution (e.g., the effect of storms on floe breakup and the relationship between sea ice concentration (SIC) and FSDs), they are limited to small areas over short periods.

Moderate-resolution Imaging Spectroradiometer (MODIS) data have been previously used in FSD studies (Toyota et al., 2016; Zhang et al., 2016; Stern et al., 2018a), but only for a short term over a few years at most. An ice floe tracker algorithm was developed (IFT, Lopez-Acosta et al., 2019) for segmenting MODIS images and tracking floes between consecutive images to evaluate sea ice-ocean interactions, but does not prioritize capturing the full FSD. We develop a new algorithm based on the work by Denton and Timmermans (2022) focusing on high identification rates of ice floes and retrieving accurate geometric properties. With 23 years of available data in the spring-to-summer transition, the MODIS optical imagery provides the potential to study interannual and decadal changes in floe properties and analyze whether there is a change in the observed FSD in the Arctic. The paper is organized as follows. First we present our algorithm for image segmentation and floe identification in optical satellite imagery in a wide range of ice concentration and melt states. We then present and validate the extracted sea ice floe properties using higher resolution data and existing datasets. We also discuss the algorithm and data limitations and uncertainty in the segmentation algorithm output. Then, the floe identification algorithm is applied to thousands of MODIS images spanning from 2000 to 2022, from March through September in the Beaufort Sea. The seasonal evolution and interannual variability of the floe sizes are presented and discussed. We conclude the study with an overview of our findings and suggestions for future directions.

## 2   Study Area and Data

The Beaufort Sea has experienced profound changes over recent decades. The end of summer sea ice area, measured as the September monthly average, is decreasing at a rate of 10-30% per decade, resulting in substantially more solar heating in the upper ocean (Timmermans and Toole, 2023). There is a loss of older and thicker multiyear ice in the Beaufort Sea (Maslanik et al., 2007; Kwok and Cunningham, 2010) and the multiyear ice edge is receding (Galley et al., 2016). Ice loss in the Beaufort Sea has influenced the Beaufort Gyre circulation and freshwater (Timmermans and Toole, 2023); we focus this study in the Beaufort Sea to capture the changes in the ice cover to further understand these important processes (Fig. 1a). We hypothesize that there is a quantifiable seasonal transition in FSD from when high concentration winter sea ice occurs to the melting fractured sea ice of summer.

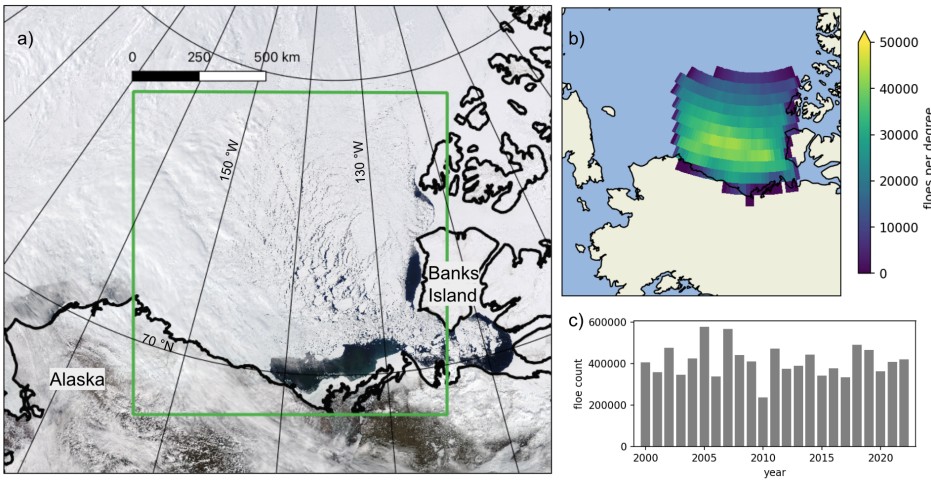

**Figure 1.** Study region. a) True Color MODIS imagery of the Beaufort Sea on 22 May 2021; the study region is outlined in a green box. b) Location density of observed floes in the study region. The 2-d histogram shows the total number of floes observed in each 1 degree x 1 degree box. c) Distribution of observed floes by year.

## 2.1 MODIS Imagery

The Moderate Resolution Imaging Spectroradiometer (MODIS) instrument is onboard two NASA satellites, Terra and Aqua, launched in 1999 and 2002, respectively. MODIS acquires data in 36 spectral bands spanning wavelengths from 0.4 to 14.4 $\mu$m with varying resolution. In this work we use the True Color imagery (Fig. 1a), a composite of Band 1 (red, 645 nm), Band 4 (green, 555 nm), and Band 3 (blue, 469 nm) (Vermote, 2015). This composite product is available at 250 m resolution. Full Arctic coverage imagery is provided twice daily outside of polar night (March through mid-October in the Beaufort Sea). The MODIS Cloud Product (MOD06/MYD06, Platnick et al. (2016)) reports calculated cloud properties for each pixel in the MODIS images and includes a cloud fraction derived from the infrared imagery. We use the cloud fraction parameter to mask out the cloudy regions in the pre-processing step of the algorithm (Section 3.1).

## 2.2 Sentinel-2 Imagery

The Sentinel-2 satellites A and B, launched in 2015 and 2017, carry the multispectral instrument, acquiring data in 13 spectral bands. Sentinel-2 imagery is captured approximately twice daily in the Arctic and available up to 20 km off the coast. The Sentinel-2 Level 1C Top of Atmosphere Reflectance product includes four bands that provide data at 10 m resolution: Band 2 (blue, centered at 492.3 nm), Band 3 (green, 558.9 nm), Band 4 (red, 664.9 nm), and Band 8 (near infrared, 832.9 nm) (Drusch et al., 2012). In this work we use the high-resolution Sentinel-2 data to understand the limitations of the lower resolution MODIS imagery.

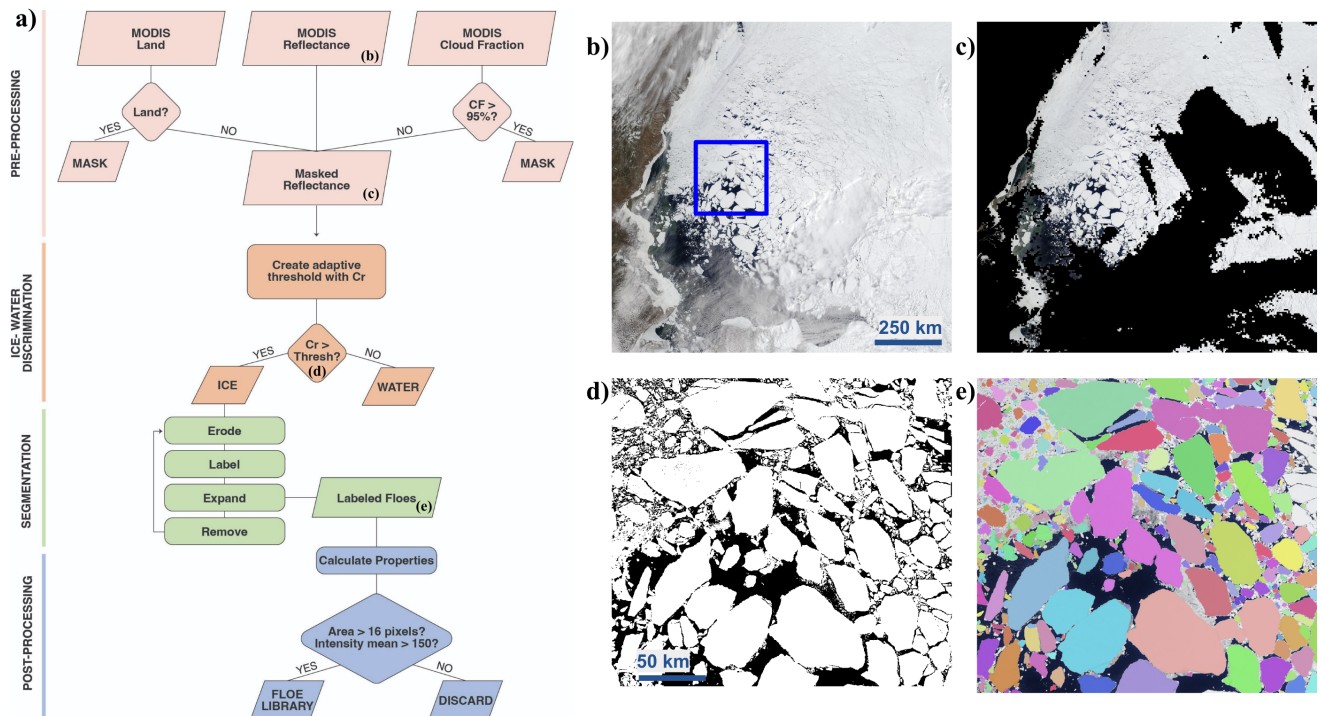

**Figure 2.** Image segmentation methodology. a) Flow diagram showing the steps of the algorithm: pre-processing (pink), ice-water discrimination (orange), segmentation (green), and post-processing (blue). b) The original MODIS true color reflectance imagery in the region of interest. The blue square shows the location of d) and e). c) The MODIS true reflectance data with a mask (black) over the land and cloudy regions. d) Ice water discrimination results showing ice (white) and water or mask (black). e) Final segmented image showing individual floes in different colors (superpositioned on the original optical image). The example MODIS image shown here is from 10 June 2017.

## 3 Image Segmentation Methodology

Paget et al. (2001) describe an erosion-expansion algorithm that erodes the boundaries of floes to separate them, and subsequently regrows them to their original shape. Denton and Timmermans (2022) and Stern et al. (2018a) introduce an iterative procedure that cycles through rounds of erosion-expansion varying the amount of erosion to identify floes of different sizes. Building on the work in Denton and Timmermans (2022), we develop a new algorithm for the identification of ice floes that capture FSDs in the MODIS dataset. We rewrite the algorithm in Python, and automate it to process thousands of images consecutively, introducing adaptive thresholds. The algorithm consists of four steps: pre-processing, ice-water discrimination, segmentation, and post-processing (Fig. 2).

### 3.1  Pre-Processing

We use the MODIS True Color Reflectance data as a starting point for our analysis. To ensure we process only clear-sky, ocean imagery, we mask the land and any cloud-covered areas. Atmospheric noise can blur the edges of the floes. Hence, we are conservative with cloud cover and use the existing MODIS cloud fraction data to eliminate areas that may be affected by clouds. We mask the areas with $\geq 95\,\%$ cloud coverage according to the MODIS cloud fraction product (Section 2.1, Fig. 2c). The presence of clouds in the Arctic in the summer is ubiquitous; on average 58% of an image is covered in opaque clouds and

not available for image processing (see discussion in Section 4 and Fig. 5a-b).

### 3.2  Ice-Water Discrimination

The images cover a large area with a range of ice conditions, i.e. from the ice edge into the pack ice region, and we cannot apply a simple threshold pixel value to distinguish the bright ice from the dark open water pixels (as is done by Denton and Timmermans (2022)). The thin leads that separate the large floes in the spring may be covered by a thin layer of ice increasing

the brightness of the lead beyond the typical brightness values of the open ocean. Similarly, the ubiquitous low-lying fog in the summer Arctic may locally brighten the appearance of open water. Considering the variable pixel values of open water, we apply an adaptive threshold to determine local values for the ice-water discrimination. At this point, the land and cloud pixels are masked and not considered in this step. We apply this method to the red channel band of the MODIS imagery which exhibits the highest contrast in pixel values. The dynamic threshold value is the weighted mean for the 399 x 399 pixel

neighborhood ($\sim$100 km x $\sim$100 km). All pixels with brightness greater (less) than the threshold are identified as ice (open water). In this way, we are able to account for varying brightness levels of open water given different ice concentration and atmospheric conditions.

### 3.3  Image Segmentation

The image segmentation step follows an erosion-expansion routine similar to that described by Paget et al. (2001) and sub-

sequent studies segmenting airborne and satellite imagery of sea ice floes (e.g.,  Denton and Timmermans, 2022; Steer et al., 2008). The input to the segmentation routine is the binary classified image created in the previous step (Section 3.2), where 1 represents sea ice and 0 is open water. The morphological erosion operation is applied to the binary image, removing pixels on the object boundaries with a diamond shaped structuring element with a radius of 1. The binary image is eroded a total of eight times in order to ensure that floes are separated. This extensive erosion removes small floes from the image. At this point, the

remaining distinct floes are tagged, and then regrown (dilated) to their original state. Any identified floes touching the image border, the land mask, or the cloud mask are removed from the binary image. The remaining tagged floes are saved in the floe library and then removed from the image, i.e., changed to 0 and the next iteration round begins for identification of smaller floes. This erosion-tag-expansion process is repeated, with fewer erosions each time, allowing for subsequently smaller floes to be identified with each iteration. In this way, sea ice floes of varying sizes are separated and identified. At the end of the

image segmentation routine, an image with each unique object labelled is produced. Over the thousands of processed images,

on average, 26% of the classified sea ice area is identified as individual floes, with remaining sections consisting of ice filaments, brash ice or pieces of ice smaller than the minimum detectable floe size. Here, we processed years 2000-2022, from day-of-year 60-274 (approximately March 1- September 30, encompassing the time of year when light levels are sufficiently high for optical imagery) and segmented 4,861 images (days), thereby identifying 9,448,563 floes.

## 3.4 Post-Processing

The following geometrical parameters are calculated for each identified floe: centroid position, floe orientation, area, perimeter, major and minor axis, circularity, and intensity mean. The major (minor) axis is the length of the major (minor) axis of an ellipse with the same normalized second central moments as the identified floe shape. The intensity mean is the average red channel value for the area of the floe. Floe orientation is defined as the angle of the major axis of the floe from polar stereographic North. Orientation values range from $-\frac{\pi}{2}$ to $\frac{\pi}{2}$. The circular standard deviation of the orientation is calculated to represent the variability of floe orientation, with low values representing floe alignment. We also calculate the circularity of a floe:

$$C = \frac{4\pi A}{P^2}, \tag{1}$$

where $A$ is the floe area, $P$ is the floe perimeter, and a circle has maximum circularity, $C = 1$.

Finally, we examine the properties of each identified object to ensure it is a floe. We examine the red channel pixel intensity mean of the floes, and discard floes with a value less than 150, an empirically determined value. Image pixels on ice floes have high red channel values, and objects with a low intensity mean may be incorrectly identified as floes; rather, low intensity values may indicate clusters of brash ice, for example. This quality assurance step results in the elimination of 86,501 floes ($<$1% of total floes).

## 3.5 Floe Size Distribution

The FSD contributes to the characterization of the ice floe field by providing a quantitative description of the ice floe area statistics. Taken together with the floe geometry properties outlined in Section 3.4, the FSD and other parameters commonly used to describe floe fields, such as the SIC and average ice thickness, allow us to study the physical processes that shape the structure and evolution of sea ice. We utilize the powerlaw Python package (Alstott et al., 2014) based on the maximum likelihood estimation power law fitting methods described by Clauset et al. (2009) and Klaus et al. (2011). The non-cumulative power law is described by:

$$p(x) = cx^{-\alpha}, \tag{2}$$

where $p(x)$ is the probability of a given instance of $x$, the chosen geometric property of the sea ice floe (we use floe area), $c$ is a normalization constant ensuring that the function integrates to 1, and $\alpha$ is the fitted parameter and slope of the power law distribution. We specify a minimum and maximum floe size value, $x_{min} = 5$ km$^2$ and $x_{max} = 300$ km$^2$, based on the range of observed floe sizes and the goodness of fit of a power law distribution to this range. Approximately 97% of the floes fall in this size range. Setting an $x_{min}$ and $x_{max}$ also allows for a finite integration of the power law. With given $x_{min}$ and $x_{max}$ values,

the constant ($c$) is given by (see Appendix A):

$$c = \frac{1 - \alpha}{x_{max}^{1-\alpha} - x_{min}^{1-\alpha}}.$$  (3)

The standard error of $\alpha$ is defined by Clauset et al. (2009) as:

$$\sigma = \frac{\alpha - 1}{\sqrt{n}} + O(1/n),$$  (4)

where $n$ is the sample size, and the higher order correction is positive. For these equations, $\alpha$ must be greater than 1. We calculate the FSD power law fit for different data sets to understand how the ice cover evolves, seasonally and annually.

## 3.6 Validation

We evaluate the consistency of the algorithm by examining the floes extracted from Aqua and Terra images on the same day. The images cover a large region and the acquisition times of Aqua and Terra are approximately two hours apart, so we can assume the images cover the same expanse of ice in same-day acquisitions. We can expect a similar power law distribution of identified floe sizes on the same day, albeit from different satellites, given that they carry the same MODIS instrument. We randomly selected 100 days from the 23-year collection of images to examine both Aqua and Terra images. We segment both images, calculate floe properties, and match floes based on centroid location. We find a correlation value of 0.99 for the matched floes areas. For each pair of images where each image has at least 50,000 km$^2$ of identified floes, we fit a power law to the floe area distribution (Section 3.5), and determine the slope ($\alpha$). We find an absolute mean difference in the Aqua and Terra $\alpha$ values of 0.009 and a standard deviation of 0.006. This suggests a strong agreement between the floes identified in the Aqua and Terra satellite imagery, confirming that the segmentation algorithm is consistently identifying floes.

To understand and quantify the limitations of the moderate-resolution imagery used by the segmentation algorithm, we apply the algorithm to higher resolution 10-m Sentinel-2 imagery. We examine spatially coincident MODIS imagery and Sentinel-2 imagery observed on the same day (Fig. 3). The extents of the two images are matched, so that the same area is analyzed. We evaluate imagery in a range of sea ice conditions: low SIC (40%) seen at the end of the summer (Fig. 3a-c), ice in the marginal ice zone in summer (70% SIC, Fig. 3d-f), and high SIC (98%) at the beginning of the melt season (Fig. 3g-i). The SIC was determined from the ice-water discrimination step in the algorithm. We pair floes identified in the coincident imagery based on centroid location and examine the corresponding floe properties. The areas of the matching 82 floes agree well, with a squared correlation of 0.99, and an absolute mean area difference of 0.18 km$^2$ (Fig. 4). The 82 floes are 25% of the Sentinel-2 floes and 21% of the MODIS floes in the observable floe area range for MODIS data ($x_{min}$ = 5 km$^2$, $x_{max}$ = 300 km$^2$). Despite a low percentage of matching floes, the identified floes in both sets of imagery are a good representation of the floe areas in each image.

We fit a power law to the FSD for all floes within the floe range in each of the images (Fig. 3c,f,i). The FSD for each pair of Sentinel-2 and MODIS images agree well. The difference in $\alpha$ values ranges from 0.06 to 0.25, with 1-sigma confidence intervals overlapping between Sentinel-2 and MODIS distributions for the low and high SIC instances (Fig. 3c and i), and 2-sigma overlap for the medium SIC example (Fig. 3f). Note that the identification of floes in the Sentinel-2 imagery was not

limited to the MODIS range of floe sizes and approximately 82% of the floes identified in the Sentinel-2 imagery are less than the determined $x_{min}$ value for the MODIS imagery. In the overlapping range of floe areas ($x_{min}$ = 5 km$^2$, $x_{max}$ = 300 km$^2$), 14% more floes are identified in the MODIS imagery compared to the Sentinel-2 imagery. The higher resolution of Sentinel-2 allows for identification of smaller floes but the segmentation algorithm performs similarly to the lower resolution imagery for larger floes. In the high sea ice concentration scenario (Fig. 3g-i) the narrow leads seen in both images make the separation of floes more challenging. The segmented floes seen in (g) and (h) look different, but the agreement of the FSD $\alpha$ indicates that the floes identified in the images are a good representation of the ice cover. We note that in high concentration regions, there is more uncertainty due to the difficulty in identifying individual floes separated by narrow leads. Despite the resolution limitations of the MODIS imagery, our validation shows that the algorithm applied to the MODIS imagery samples the floe sizes sufficiently to produce an accurate FSD $\alpha$ value, and that the floes identified in both images have highly correlated floe areas.

Finally, we apply a bootstrap approach to quantify uncertainty in the derived parameters. We begin by randomly selecting 100 segmented MODIS images (corresponding to 100 different days), where each image has at least 50,000 km$^2$ of identified floes. We then create 1,000 bootstrap datasets of floes for each of the 100 images, where each bootstrapped sample has the same number of identified floes ($N$) as the original image. This is done by randomly selecting $N$ floes from an image, such that after each floe is selected, it is returned to the original image (i.e., sampling with replacement). We then calculate the power law distributions of the bootstrap datasets, and the standard deviation of the $\alpha$ values over the 1,000 bootstrap datasets for each of the 100 images. The standard deviation of $\alpha$ generated from the bootstrapping of an image is on average, 0.024, ranging from 0.007 to 0.08. The standard deviation increases as the $\alpha$ value increases, indicating more uncertainty.

## 3.7  Data and Algorithm Limitations

We note that there are limitations to our analysis due to the moderate resolution of the imagery. The separation of floes requires openings between floes (such as leads) that are at least the size of an image pixel (250 m), and typically multiple pixels are required to fully resolve a lead. This results in multiple floes being considered as a single floe (examples can be seen in Figure 3g). This phenomenon is especially prevalent in early spring, when floes are still tightly packed and have not experienced lateral melt and floe divergence. In addition, larger floes are more likely to intersect the image border. Floes that intersect with the image border are removed from the identified floes as the properties are not correct, thus large floes are preferentially eliminated. For these reasons, we limit the range of the power law fit, and do not make conclusions on the power law distribution of floes > 300 km$^2$. Nonetheless, the error associated with the FSD is largest when the SIC is high in early spring.

We chose to analyze MODIS imagery due to its long record and consistent coverage, but other higher resolution imagery is required to examine the FSD for floes smaller than 5 km$^2$. Other studies have examined FSD in Synthetic Aperture Radar (SAR) imagery which eliminates the need for clear-sky conditions as SAR is not sensitive to clouds. However, SAR data are not as widely available for long-term applications, and have other limitations and complications, such as speckle, granular noise, and ambiguous returns when meltwater is present on the ice surface.

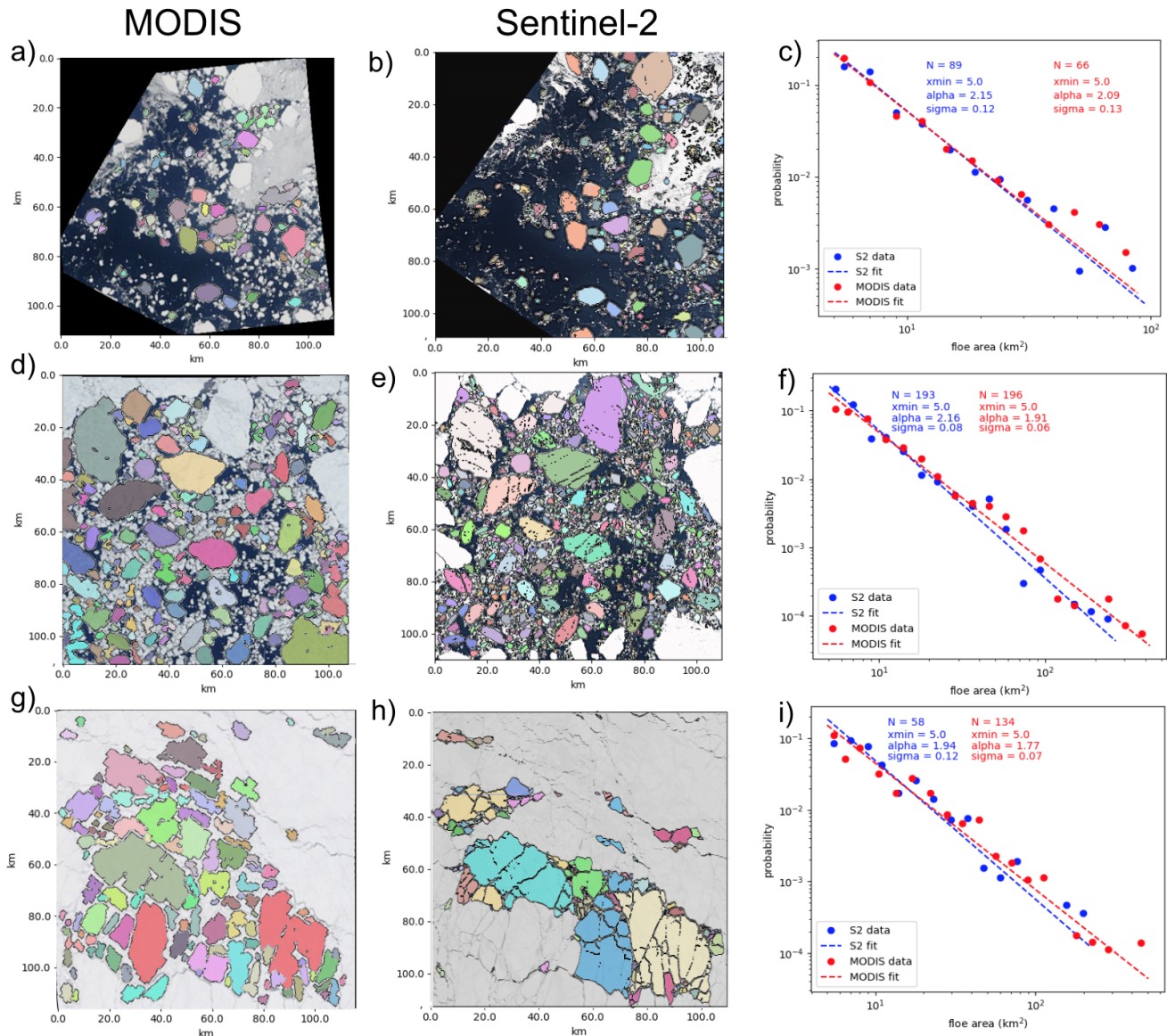

**Figure 3.** Validation of MODIS imagery segmentation with higher-resolution (10 m) Sentinel-2 imagery. Each row shows spatially coincident MODIS (first column) and Sentinel-2 (second column) imagery captured on the same day. The segmentation algorithm is applied to each of these images and the individual identified floes are colored in the image. The third column shows the PDF of the FSD for Sentinel-2 (blue) and MODIS (red) images, with the best fit power law shown as the dashed line. The three rows are for imagery captured on 4 September, 2019 (40% SIC), 12 June, 2020 (70% SIC), and 14 May, 2021 (98% SIC).

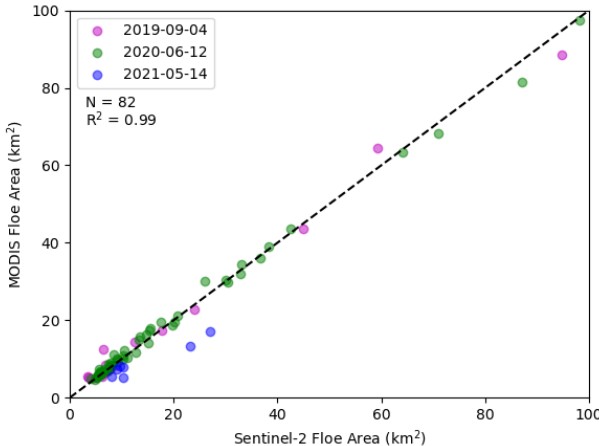

**Figure 4.** Area comparison of identifications in MODIS and Sentinel-2 imagery 3. The 82 floes identified in both data sets are shown with the dot color corresponding to the date of the images as indicated in the legend.

## 4 Spring to Summer Transition of Floe Characteristics

After processing 4,861 images we analyze the basic statistics of the classification and segmentation routine. Over 9.4 million floes were identified; the distribution of their locations is shown in Fig. 1b; note that these are not 9.4 unique floes as each image is taken as an independent observation. The MODIS cloud fraction in the study region is consistent with the Arctic wide pattern of highest cloud fraction in the summer and fall (Fig. 5) (Schweiger, 2004). Although this study is not focused on atmospheric trends, we do not find a significant long term trend in the cloud fraction over our study region and period (Fig. 5b). We note that our analysis of only cloud-free area is limited due to a high percentage of cloud cover obscuring the sea ice in optical imagery. As expected, the total classified sea ice area and total identified floe area in the imagery decreases throughout the summer as the ice melts (Fig. 5c, orange, blue). The largest difference between the ice area (orange) and identified floe area (blue) is in the spring. There are a number of reasons why the majority of the ice cover is not able to be segmented into floes in the spring. The floes in the spring are larger (Fig. 6a) and therefore are more likely to intersect with the border of an image, and eliminated to ensure only floes fully captured by the imagery are used. Also, the ice floes are tightly packed in the spring and the MODIS imagery does not resolve small leads (Fig. 3h), and thus cannot separate floes as well compared to the summer where floes are separated by larger areas of open water (Fig. 3e). The number of identified floes (Fig. 5c, gray) is not correlated with the total area of observed floes (Fig. 5c, orange). More floes are identified as the ice separates in the summer, but as the ice melts and advects out of the study area, there is less ice and fewer floes identified. There are no significant trends in the ice and floe areas or the number of floes identified over the 23 years of observations (Fig. 1c and 5d).

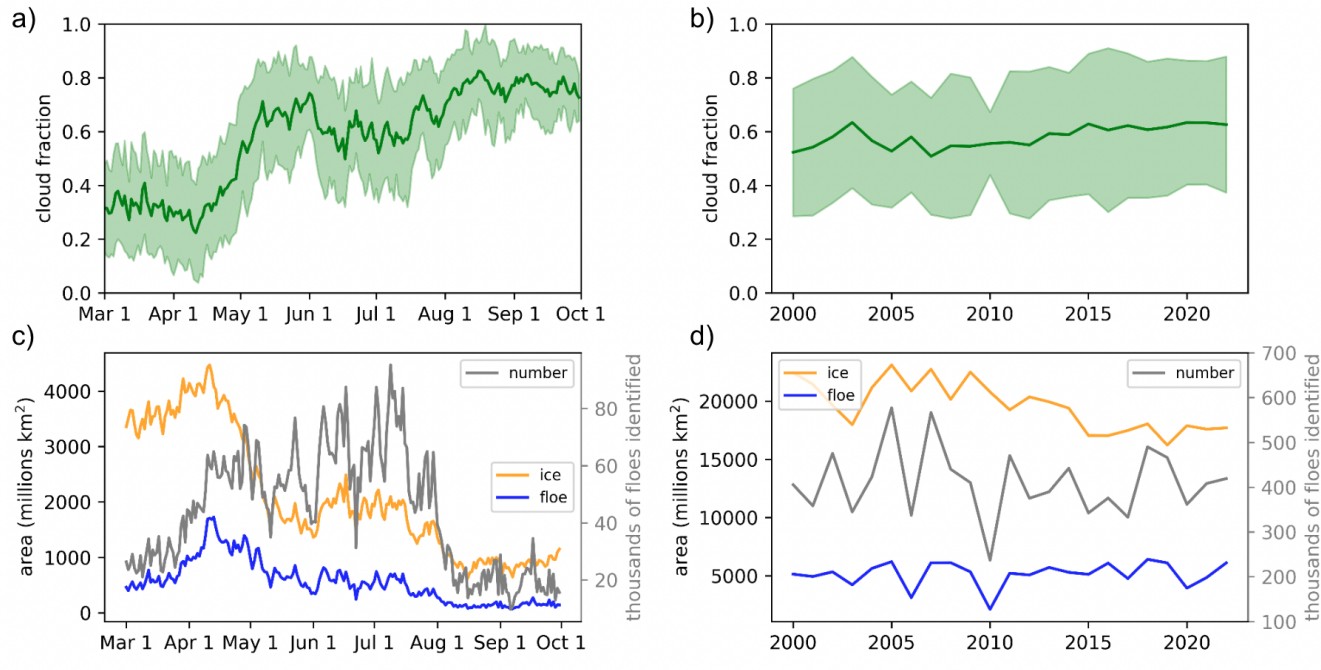

**Figure 5.** Image statistics. a) and b) show cloud fraction statistics from the cloud mask. c) and d) show the total ice area identified from the ice water discrimination (orange) and the total area of the identified ice floes (blue). The gray line shows the number of ice floes identified. a) and c) show statistics over the season, and b) and d) show statistics over the years.

## 4.1 Floe Area

In the spring, the Beaufort Sea has a high ice concentration consisting of large floes with rectilinear fractures. During the summer, as the ice edge recedes from the Alaskan and Canadian shorelines, the ice cover transitions to a dynamic collection of randomly oriented floes among brash ice. We analyze floe characteristics from March 1 through September 30 in the MODIS imagery and observe patterns corresponding to this transition. The mean floe area first increases from 32 $km^2$ on 6 March to the seasonal high of 36 $km^2$ on 10 April (Fig. 6a, blue). Mean floe area then decreases throughout the summer, plateauing

in August around 21 $km^2$ and then increasing to reach a mean area of 23 $km^2$ by the end of September. The mean floe area closely follows the 75th percentile floe area (Fig. 6a, dark blue). The median floe area is consistently lower than the mean floe area (Fig. 6b, orange), as the distribution is positively skewed with many small floes. Because of the high skewness of the distribution, the standard deviation is large (38.9 $km^2$ on average) as it is influenced by floes much larger than the mean floe area. The median floe size exhibits a similar pattern as the mean, with a maximum median value of 16 $km^2$ on 10 April,

decreasing to 11.5 $km^2$ 30 September. We also see the largest floes in areas of highest SIC (Fig. 7b). Large ice floes and high SIC exist in the early spring. Floes that exist in areas of low ice concentration are more likely to experience the effect of waves, and may break up into smaller floes due to wave fracture (Squire et al., 1995).

We note that the magnitude of the median and the mean floe area values are sensitive to changes in the $x_{min}$ and $x_{max}$ values. However, the pattern of evolving floe sizes is related to the changing sea ice cover. The mean (expected value) and median (where the cumulative distribution function is 0.5) of the fitted power law can be expressed analytically (see Appendix A2 and A3). These values, with $x_{min}$ and $x_{max}$ set, are functions of the evolving $\alpha$. We show the analytical mean and median as dot-dashed lines in Fig. 6a-b, respectively, which indicate that the analytical solutions are consistently less than the observed mean and median floe areas. This is because the power law fit is dominated by small floes, as there are more of them. Thus, there is greater error in the fit associated with the large observed floes.

## 4.2 Power Law Fit to Floe Size Distribution

We fit a power law to the collection of floes grouped in a 10-day running window. The value of $\alpha$ (slope of the power law distribution) of all floes identified is 1.85, but ranges from 1.74 to 2.0 throughout the spring-to-summer period (Fig. 6c). The standard deviations and the confidence intervals of all power law fits are less than 0.01. The power law $\alpha$ is inversely correlated to the mean floe area; the slope value decreases from the beginning of March to the minimum on 9 April, then increases throughout the summer to a maximum on 12 August, and slowly begins to decrease again for the remainder of the summer. The inverse relationship to mean floe area is expected (see Appendix A2), as a higher frequency of small floes will decrease the mean floe area and increase the slope of the FSD fit. We find the magnitude and seasonal trends of $\alpha$ consistent with previous studies.

Stern et al. (2018a) examined 116 MODIS images in 2013 and 2014 and fit a power law to floes ranging from 2 to 30 km in size, which is approximately equivalent to 2.64 to 594 km$^2$ (Rothrock and Thorndike, 1984). To compare results, we must account for the fact that we have fit a power law to the floe area, while Stern et al. (2018a) find a power law slope ($\alpha_n$) using the mean caliper diameter. This value is related to our reported $\alpha$ as (see e.g., Denton and Timmermans (2022)):

$$\alpha_n = 2\alpha - 1. \tag{5}$$

Stern et al. (2018a) found $\alpha_n$ ($\alpha$) was approximately 2.0 (1.5) in May, increased to about 2.9 (1.95) in July, and then decreased to about 2.2 (1.6) by October. We find comparable slopes and observe a similar evolution with $\alpha$ about 1.75 in May, a maximum $\alpha$ in August of 2.0, and then decreasing slightly in September (Fig. 6c). The small differences in values are likely due to location and time of the observed floes. Denton and Timmermans (2022) analyzed smaller floes (50 m$^2$ to 5 km$^2$) in the Canada Basin and found a seasonal trend in FSD with $\alpha$ ranging from 1.65 to 2.03.

The FSD trend is also consistent with previous studies that have taken a Lagrangian approach, tracking the same ice throughout the summer. During the Surface Heat Budget of the Arctic (SHEBA) campaign in 1998 in the Beaufort and Chukchi Seas aerial photography was collected in the proximity of the ship. Perovich and Jones (2014) found $\alpha$ values increased through the summer, reached a maximum on 10 August, and subsequently decreased into September as small floes froze and fused into larger floes (Perovich and Jones, 2014). Hwang et al. (2017) observed the FSD from satellite SAR imagery tracking four buoys in the Beaufort Sea in 2014 and capturing images of the same ice, finding an increase in FSD $\alpha$ values from July through August with enhanced floe breakup linked to wind events.

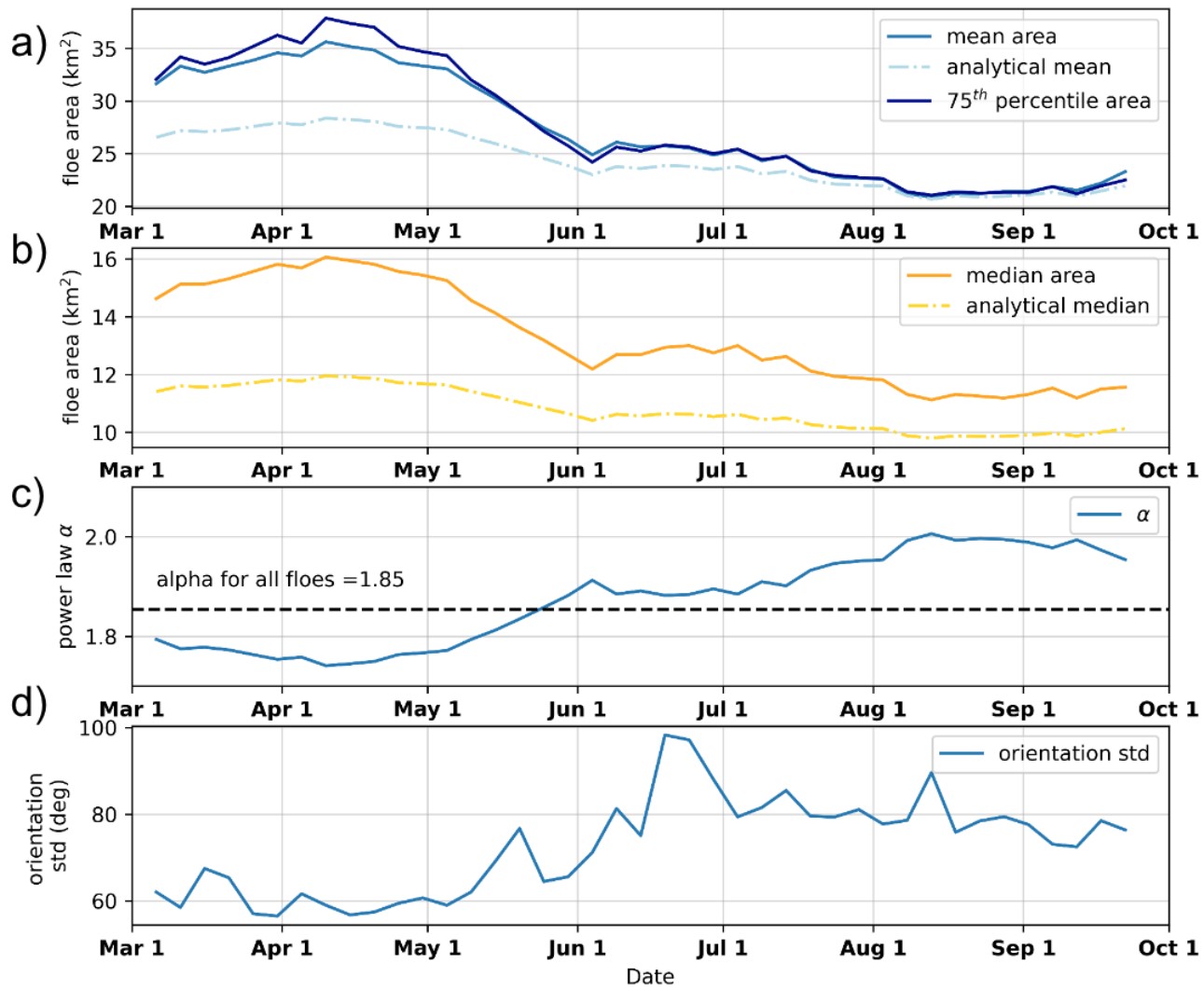

**Figure 6.** March 1 through September 30 evolution of floe properties. The properties were aggregated into a 10-day running window and sampled every 5 days. a) mean floe area for the observed (solid) and analytical (dashed) values (See Appendix A2). The 75th percentile of the floe area is also shown (dark blue). b) median floe area for the observed (solid) and analytical (dashed) values (See Appendix A3). c) the power law slope ($\alpha$). d) standard deviation of the orientation of floes for all floes.

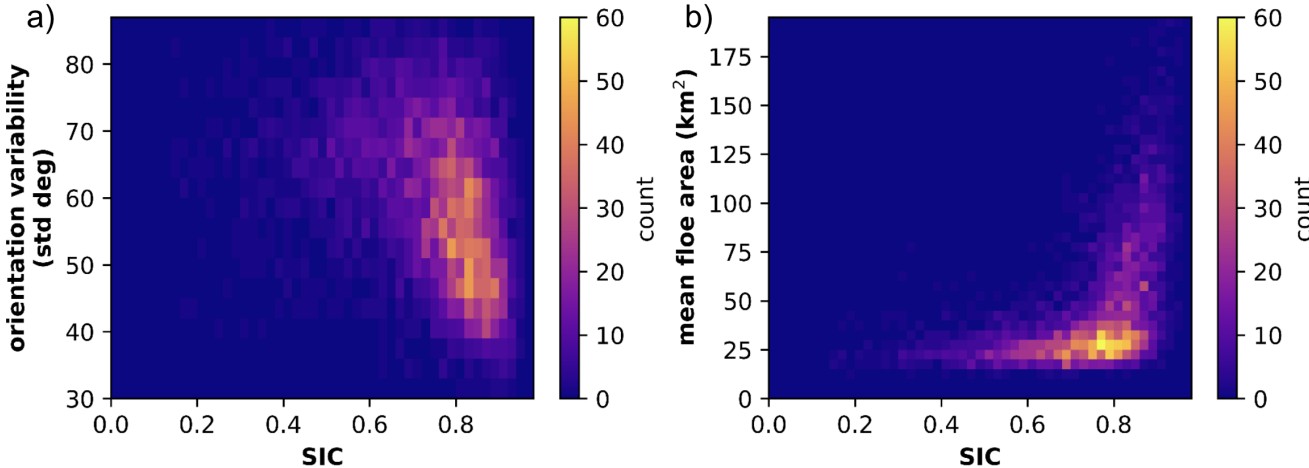

**Figure 7.** Observed floe properties compared to sea ice concentration (SIC). a) Standard deviation of the orientation of floes on a given day compared to the image SIC. b) Mean floe area compared to SIC on a given day. The colorbar indicates total image count.

We examine the FSD power law slope over the 23-year MODIS record to quantify the interannual variability and discern any decadal trends. As the FSD exhibits seasonal variability we look at the monthly FSD power law slopes and compare over the years. We see no significant trends in the monthly FSD slopes over the 23-year period (Fig. 8). The error on all calculated $\alpha$ values is < 0.01 as determined by 1,000 bootstrap samples to calculate a 95% confidence interval, and the standard error
calculated as in Equation 4. The month of September exhibits the most variability with $\alpha$ values ranging from 1.85 to 2.17 (Fig. 6c, and 8). One may expect a trend towards larger $\alpha$ values (steeper FSD power law slopes) as the Beaufort Sea ice cover exhibits earlier retreat (Fetterer et al., 2017) and a transition to first year ice that is more susceptible to fracture (Galley et al., 2016), however this is not exhibited in our data. This lack of a significant decadal trend in FSD slope may be due to the large study region that simultaneously contains pack ice and open water, or, because MODIS cannot resolve floes smaller than 5
280 km$^2$ in area, there may be many more small floes in recent years that are not identified in the MODIS data.

### 4.3 Floe Orientation

In the late winter and spring, the Beaufort Sea has high SIC. As external wind stresses are applied to the ice pack, the ice pack experiences strain, leading to fracture in a preferential direction that depends on the orientation of the force relative to the coast (Lewis and Hutchings, 2019; Jewell et al., 2023). When we examine the newly fractured floes during the spring,
we find low variability in their orientation within the 10-day window (Fig. 6d). This effect is especially noticeable in areas of high ice concentration, where the ice movement and readjustment to external forces is limited by the surrounding ice and thus lower standard deviation of orientation with greater SIC (Fig. 7a). The small fractures, or cracks, grow into leads that can be seen in the MODIS imagery (e.g., Fig. 3g). As the summer progresses, SIC decreases and ice floes break up and disperse, resulting in a decreasing average floe area (Fig. 6a) and an increasing standard deviation of floe orientations with lower sea

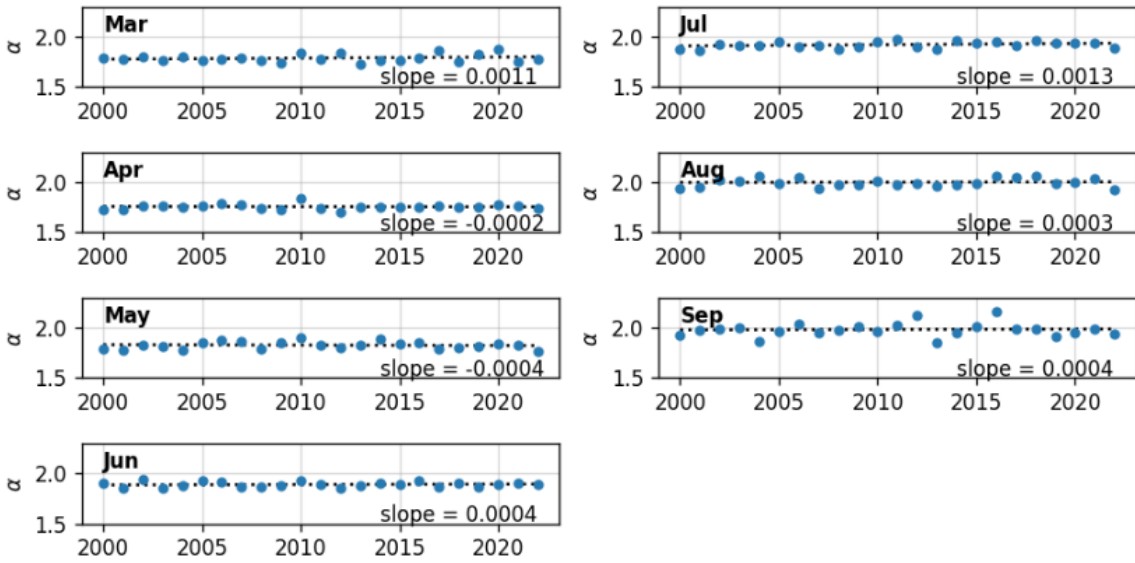

**Figure 8.** $\alpha$ values of a power law fitted FSD for each month of analysis. Each strip shows the $\alpha$ value for all floes identified in that month over the 23-year analysis. The dotted line is the line of best fit, the slope of which is printed in the bottom right hand corner of each strip.

ice concentrations (Figs. 6d and 7). We also examine the floes with the highest eccentricity (>75th percentile) which is the ratio of the major axis to the minor axis. This subset of floes exhibits the same but less extreme trend of an increasing standard deviation of orientation through the summer. The orientations of the floes are an indication of the stresses that have caused ice breakup and can provide insight into the structural properties of the ice pack to inform how it may respond to future stresses. When ice floes can rotate with minimal to no interaction with other floes, their rotation rates can be related to ocean vorticity (Manucharyan et al., 2022).

## 5 Conclusions

The algorithm developed in this study establishes the ability to derive meaningful floe size information from the longest daily global satellite observation record of Earth to date. The MODIS dataset has Arctic wide coverage and spans 23 years. The expansive dataset combined with the modified segmentation algorithm presented here allows for continued study of ice conditions and characteristics in the spring and summer. To validate the feasibility of using moderate resolution imagery, the identified floes are validated visually and compared with floes identified in higher-resolution Sentinel-2 imagery. We demonstrate that in the specified range of floes with area greater than 5 km$^2$, the segmentation algorithm performs equally well on images of moderate resolution (MODIS) and high resolution (Sentinel-2). Although the algorithm is able to identify smaller floes in the Sentinel-2 imagery, the floe sizes retrieved and the FSD agree well between the Sentinel-2 and MODIS imagery. This allows

us to confidently apply this systematic method to analyze thousands of images over 23 years covering a wide range of ice conditions.

We examine the seasonal evolution of the FSD and aspects of the floe geometry in the Beaufort Sea from mid-April through early August. We find a decrease in the mean and median floe size, increasing $\alpha$ (steepening power law slope), and an increase in the variability in the orientation of the floes. As the sea ice cover appears and behaves significantly differently depending on the time of year, it is essential to use floe characteristics from the specific time period of interest when evaluating or tuning models. While no significant decadal trends were observed in the monthly FSD over the 23-year period, future work considering smaller spatial domains may be necessary to investigate interannual and decadal variability in detail.

Expanding analysis to new areas of the marginal ice zone may show regional differences in floe characteristics and timing of floe break up in the summer. Combining this new information with existing satellite measurements (e.g., ice drift, ice type, ice thickness) can provide further insights into the behavior of the ice pack. We chose to analyze MODIS imagery due to its long record and consistent coverage, but other higher resolution imagery is required to analyze how the FSD holds for small floes. It also may be of interest to examine floe evolution throughout the winter, which would require active sensors that can produce images when there is no sun illumination available. Further work will include incorporating this image segmentation algorithm into the pipeline of the Ice Floe Tracker algorithm (IFT, Lopez-Acosta et al., 2019). This routine tracks floes with similar characteristics between consecutive MODIS images and can thus determine ice velocities and rotation rates, inferring ocean dynamics in regions that are otherwise under-observed. With a new segmentation algorithm able to identify floes in a wider range of sizes, we can expand the IFT output to also include FSD and uncover more information about the underlying ocean.

*Code and data availability.* Ice floe segmentation algorithm code is available at: https://github.com/WilhelmusLab/Segmentation_EB. This algorithm will be incorporated into the Ice Floe Tracker algorithm pipeline here: https://github.com/WilhelmusLab/IceFloeTracker.jl. Results from the segmentation algorithm are archived on Zenodo, DOI: https://doi.org/10.5281/zenodo.11553700.

*Author contributions.* MMW conceived of the study. EB wrote the algorithm and applied it to the imagery. MLT and LC supported the analysis of the results. All authors contributed to the writing of the paper.

*Competing interests.* We declare that no competing interests are present.

*Acknowledgements.* EB, MLT, and MMW were supported by the Office of Naval Research (ONR) Arctic Program (N00014-20-1-2753, N00014-22-1-2741, and N00014-22-1-2722) and the ONR Multidisciplinary University Research Initiatives Program (N00014-23-1-2014).

We thank Daniel Watkins for the internal manuscript reviews, and Harding Coughter for assistance in the graphic design of the figures. We thank the two anonymous reviewers for their helpful comments and corrections.

## Appendix A: Mathematical Derivations for Power Law Floe Size Distribution

### A1   Calculation of the Power Law Constant

To solve for $c$ in the power law distribution (Equation 2), we integrate the power law from $x_{min}$ to $x_{max}$, noting that the sum of all probabilities in the range is = 1

$$\int_{x_{min}}^{x_{max}} cx^{-\alpha}\,dx = \frac{-cx^{1-\alpha}}{\alpha-1}\Big|_{x_{min}}^{x_{max}} = -\frac{cx_{max}^{1-\alpha}}{\alpha-1} + \frac{cx_{min}^{1-\alpha}}{\alpha-1} = 1, \tag{A1}$$

where $\alpha > 1$.

This yields

$$c = \frac{1-\alpha}{x_{max}^{1-\alpha} - x_{min}^{1-\alpha}}. \tag{A2}$$

### A2   Analytical Mean

The expected value of a distribution is:

$$E = \int_{x_{min}}^{x_{max}} xp(x)\,dx. \tag{A3}$$

With Equation 2, this is

$$E = \int_{x_{min}}^{x_{max}} cx^{1-\alpha}\,dx, \tag{A4}$$

where $c$ is given by Equation (A2). Integrating yields

$$E = \frac{c}{2-\alpha}\left(x_{max}^{2-\alpha} - x_{min}^{2-\alpha}\right), \tag{A5}$$

where $E$ depends only on $\alpha$ because $x_{min}$ and $x_{max}$ are set as 5 km$^2$ and 300 km$^2$, respectively. The full equation, substituting $c$ from Equation (A2), is

$$E = \frac{1-\alpha}{2-\alpha}\frac{x_{max}^{2-\alpha} - x_{min}^{2-\alpha}}{x_{max}^{1-\alpha} - x_{min}^{1-\alpha}}, \tag{A6}$$

for $\alpha \neq 2$.

### A3   Analytical Median

The median of a distribution $x_o$ is found where the cumulative distribution is equal to 0.5. That is,

$$\int_{x_{min}}^{x_o} p(x)\,dx = \frac{c}{1-\alpha}x^{1-\alpha}\Big|_{x_{min}}^{x_o} = 0.5. \tag{A7}$$

Solving for $x_o$ yields

$$x_o = \left[ x_{min}^{1-\alpha} + \frac{1-\alpha}{2c} \right]^{\frac{1}{1-\alpha}} , \tag{A8}$$

which, with Equation (A2), yields

$$x_o = \left[ \frac{1}{2} (x_{min}^{1-\alpha} + x_{max}^{1-\alpha}) \right]^{\frac{1}{1-\alpha}} . \tag{A9}$$

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
