# Peer review of "Seasonal Evolution of the Sea Ice Floe Size Distribution in the Beaufort Sea from Two Decades of MODIS Data"

_EGUsphere, 2024_

## Referee Comment (RC2)

**Review of the manuscript "Seasonal evolution of the sea ice floe size distribution from two decades of MODIS data" by Buckly et al.**

The authors presented a study on sea ice floe size distribution in the Beaufort Sea based on the long-term MODIS dataset.

The method of ice floe segmentation is primarily based on the previously developed method by Paget et al. (2001). MODIS-derived ice floe segmentation was compared with the Sentinel-2 data based on three cases. They applied the method to 4,861 images of MODIS, identifying more than 9.4 million floes over 23 years. By analyzing the FSD in the Beaufort Sea over a long period, they unfolded the seasonal variability, including the decreasing mean floe area and increasing FSD power law slope as the summer progresses.

Some major comments and specific comments are below for reference.

**Major Comments**

1. Obviously, the title of the manuscript is too wide, and the authors only conducted analysis in the Beaufort Sea.

2. The dataset is not well described. For instance, are all four bands of the S2 data used to segment ice floes?

3. The methodology is too brief to be understood. Is the method exactly the same as that described by Paget et al. (2001)? Are any improvements achieved?

4. Validation of the MODIS-derived sea ice floes is not convincing. On the one hand, only by presenting only three cases for nearly 5000 MODIS images seems to be inadequate. I would suggest that the authors add comparison experiments to provide a more robust argument. On the other hand, the current comparison itself is not convincing. As the three cases shown in Fig.3, there are obvious distinctions (e.g., the N values in the figure) between the two datasets. The MODIS data have a spatial resolution of 250 m, which is much larger than the S2 data of 10 m. Therefore, a threshold should be set to exclude those floes that MODIS cannot observe at all due to its coarse spatial resolution. Moreover, the validation shown in subsection 3.6 fails to "perform equally well". These ice

floes, which are not segmented by MODIS, not only include the clustered loose ice but also some pack ice. Such a discrepancy is too visible to be explained by the different data resolution. The proposed method probably has limitations in terms of accuracy (however, it is hard to judge because the method is described too briefly). It is possible that some steps, such as the use of adaptive thresholds, excessive morphological processing, or the direct discarding of low-intensity floes, lead to unreasonable results.

5. The title of section 4 should be narrowed down. I think that many relevant studies have been conducted with the FSD in the Arctic. Why not compare some previously derived FSD with the present results for further verification of its accuracy? Additionally, the statistical charts should be improved. For Fig. 5, it is fine to apply the "10-day running window", but one should also consider quantitative results such as scatters, boxplots, or upper and lower significance intervals to give the reader a clearer view of the author's raw statistical results. It would be better to try seasonal statistics, which might work better for the author's dataset (as too many MODIS data are excluded from analysis due to the cloud effect).

**Specific Comments**

P2 L25 "floe size distribution": The full name has already been presented. Similar issues occurred several times in the manuscript, including "SIC", "SAR", and "MODIS". Please revise them.

P2 L26 "see Stern et al. (2018b) for a comprehensive list of FSD studies": Even without a systematic review, a proper overview and summary of state of the art should be briefly presented here.

P2 L27-28 "... from radar imagery ... high-resolution imagery ....": The summary of research types is weird. Isn't a SAR image a high-resolution image? Please rewrite this sentence to provide a detailed review of these studies.

P2 L29-L31 "These studies have advanced our knowledge of seasonal evolution of the FSD...": The authors' review does not serve to summarise this knowledge.

P2 L35 "(Lopez Acosta et al., 2019) demonstrate..." -> "Lopez Acosta et al. (2019) demonstrate...". Please also note other similar citations.

P3 Fig.1 (c): Rather than showing the annual average ice floe numbers here, I'd be more interested in first finding out the annual use of MODIS data. In particular, long-term statistics need to know the amount of cloud-contaminated data for each year.

P4 Fig.2: The authors presented a very clear case of pack ice. In addition, I would also like to see the algorithm's adaptability to high SIC, melting ice, MIZ, ice-water mixing, etc. After all, the authors aimed to focus on the phenomena related to the transition, which implies a rather complex ice condition.

P5 L90 "400-pixel (100 km) neighbourhood of a pixel subtracted by a constant": What is the constant? The 400-pixel neighbourhood is a relatively large region. However, the masked MODIS contains many NAN values. How did the author choose the threshold at the edge of the NAN?

Additionally, it would be better for the authors to argue for the rationality and specific benefits of adaptive thresholds where appropriate.

P5 3.3 section: Since erosion-expansion is an important step, it would be better to show the effect before and after morphological processing using the Fig.2 case.

And, what is the necessity of performing multiple iterations? Actually, in my opinion, besides the fact that it does improve the visual effect, too much morphological processing may lead to losing the original sea ice features.

P5 L103-L105 "On average, 26% of the classified sea ice area is identified as individual floes": Average of what?

P5 L111 "…the variation of the floe orientation": Which floes are the authors using circularity std to compare their variation? Are they ice floes that are tracked between time-series images, or are they all ice floes in the same image?

P6 L116-119: In fact, the low-intensity ice may be an important ice condition as well (especially in the transition, where it may represent a melting scenario). However, the author removed them outright, which would cause the subsequent results, especially for power law distributions, to be different from previous results.

How can the authors justify this proposed step? Also, what are the units of 150? Can the authors prove <150 to be so-called brash ice rather than other types of sea ice (e.g. grey ice, melting ice)?

P6 L122-123 "that complements other parameters commonly used to describe ice floe fields, such as the sea ice concentration (SIC) and average ice thickness": I would suggest deleting this sentence.

P7 3.6 section: When using S2 images for segmentation, is there also a step to remove the <150 intensity? And, are the other parameter settings exactly the same (number of erosions and the calculation of adaptive thresholds)? What level of S2 data is used? These details may lead to a different adaptation of the ice floe segmentation algorithm to S2 data.

Fig 3&4: As I mentioned above, I don't think the comparison presents a good result.

What is the "$x_{min}$"? Also, please present clearer quantitative results in the text.

L149-151 "The areas of the matching 82 floes agree very well, with a correlation of 0.99, and an absolute mean area difference of 0.18 km2 (Figure 4)": The authors only compared the 81 floes identified by both. I don't think this comparison is fair. The high-resolution S2 results should be treated as a reference, and all the S2-segmented floes should be compared to MODIS results. Obviously, the MODIS results have gross omissions.

P7 L153-154 "...14% more floes are identified in the MODIS imagery compared to the Sentinel-2 imagery.": How can the author tell the 14%? I can only find that MODIS significantly underestimated the floe numbers.

P7 L163 "We create 1000 bootstrap samples with replacement": I can't follow the author's point of this step.

P7 L171 " SAR data is not as widely available": This sentence is misleading. If it refers specifically to long-term applications, SAR does have limitations. "other limitations or complications, such as speckle or granular noise". It is not correct. It is just the authors do not know how to deal with the valuable dataset.

P8 L181-182: "…median floe size exhibits a similar pattern as the mean, with a

maximum median value of 17 km2 on 10 April...": The variations in the median are not visible at all, and this image should be modified.

P11 L210 "... Stern et al. (2018a) ...": Stern (2018a) included the analysis of small ice floes (<5km2), right? So, it can be different from the authors' results.

In fact, in section 3.6, the alpha estimated from the MODIS is also lower than the S2 result. It is suggested that the proposed algorithm is supposed to have some limitations. I would suggest that the authors explain such reasons in detail.

P12 L223 "…we find a low variability in their orientation (Fig. 5c)": Low variability? It seems to me that Fig. 5(c) displays a clear increasing trend. Doesn't this suggest that as the sea ice melts, the direction of the ice floes becomes more cluttered?

P12 L223-224 "This effect is especially noticeable in areas of high ice concentration, where the ice movement and readjustment to external forces is limited by the surrounding ice ": It's hard to follow. If the authors mean that there is a decreasing tendency for std as SIC increases, it should be expressed more clearly. Otherwise, the most obvious phenomenon on this graph should be the very wide range of std variation for SIC > 0.7.

P12 L227 "large rectilinear floes": This is less common usage, so please confirm.

---

## Author Comment (AC1)

Reviewer 1:

The authors processed 4861 MODIS images of sea ice in the Beaufort Sea spanning the months of March through September in the years 2000 through 2022, identifying more than 9 million ice floes. They then constructed and analyzed the sea-ice floe size distribution (FSD). They found that the FSD follows a power law with an exponent that varies seasonally. They also calculated the mean floe size and the orientation of floes.

Previous studies of the FSD have typically been based on the analysis of a handful of images, or at most a few hundred. The amount of imagery analyzed in this work exceeds all previous studies by a factor of 15 or more, and for that reason alone it is worthy of publication. But regarding the seasonal evolution of the FSD, the authors did not make a careful comparison of their results with previous studies, and there are some mathematical issues to resolve.

**Response:** We are grateful to the reviewer for a thorough review of our manuscript. We have revised the manuscript per the reviewer's constructive suggestions, which we believe has greatly improved the clarity of the presentation. We have addressed the stated concerns and made the necessary changes to the manuscript (shown in blue font). Significant changes were made to Figure 5 (now Figure 6) per the reviewer's comments and added a new appendix to show the analytical derivation of the observed relationships in our dataset. The specific revisions are described in the point responses below:

Lines 89-90. "The dynamic threshold value is the weighted mean for the 400-pixel (100 km) neighborhood of a pixel subtracted by a constant." A few questions:

Thank you for your comment. The text has been revised to clarify:

"The dynamic threshold value is the weighted mean for the 399 pixel x 399 pixel neighborhood (~100 km x ~100 km)."

1. A neighborhood is an area. Is the neighborhood 400 x 400 pixels (100 x 100 km), or is it 20 x 20 pixels (400 total, 10 x 10 km)?

Correct. In this case, it is 399 pixels x 399 pixels (centered around the pixel of interest).

2. Is the neighborhood a square or some other shape?

Yes, square.

3. What does "subtracted by a constant" mean? What constant?

Thank you for catching this detail. We do not subtract by a constant, so this has now been deleted. This was the description of the python function we used. But the default constant is zero.

Lines 161-167. I do not understand the bootstrapping procedure. Samples of (what?) are drawn from (what?).

Thank you for your feedback. We are referring to sample of floes in an image, drawn from all the floes in an image with replacement. Text has been modified for clarity:

Finally, we apply a bootstrap approach to quantify uncertainty in the derived parameters. We randomly select 100 days of segmented MODIS imagery with at least 50,000 km^2 of identified floes, and examine the uncertainty in the derived FSD for each of those days. We create 1000 bootstrap samples (of floes in the image) with replacement. Each bootstrapped sample is the length of the original dataset (the number of identified floes in the image), and we calculate the power law distribution of the sampled floe areas. We then calculate the standard deviation of the alpha values for each image. The standard deviation of alpha generated from the bootstrapping of an image is on average, 0.024, ranging from 0.007 to 0.08. The standard deviation increases as the alpha value increases, indicating more uncertainty.

Lines 178-182. These sentences discuss the mean and median floe areas as shown in Figure 5(a). It should be noted that for a power-law distribution, the mean and the median are directly proportional to the minimum floe area, Amin, which is 5 km^2 in the present work. The power-law pdf with exponent -alpha is:

f(A) = ((alpha-1)/Amin) * (A/Amin)^(-alpha)

The integral of f(A) from Amin to infinity is 1.

The mean floe area is:

Amean = Amin * (alpha-1)/(alpha-2)

The median floe area is:

Amed = Amin * (2^(1/(alpha-1)))

The point is this: The mean and median floe areas are completely dependent upon the resolution of the measuring device. If the resolution of the sensor were to allow the detection of floes 10 times smaller, the mean and median floe areas would be 10 times smaller (provided the data continued to follow the same power law). The mean and median floe areas reported here do not reflect any kind of objective reality but rather the limitations of the sensor. They cannot be meaningfully compared to the mean and median floe areas detected by sensors with different resolutions. It's fine to calculate and plot the mean and median floe areas, and to note their variation over time, but their direct proportionality to the resolution (Amin) of the detection mechanism should be noted, if in fact the floe areas are power-law distributed.

Thank you for the comment and the detailed derivation of these equations. The mean and median values, while dependent on the Amin value (as pointed out), are also a function of alpha, which we show has a seasonality. Therefore, showing the evolution of the mean and median floe size is

an interesting result. Throughout our analysis, we have held the xmin and xmax values constant and changes in the mean and median are a result of changes in alpha. When comparing different sensors, we held xmin and xmax fixed to ensure comparability of the calculated FSDs. We have added a discussion on the relationship between these parameters (section 4.1 second paragraph) and have described the choice of xmin and xmax values in more detail (section 3.5). Thank you for pointing this out, we believe the discussion has improved with this suggestion.

The previous comment raises a further difficulty. For a power-law distribution with exponent -alpha, the mean does not exist when alpha LE 2. In other words, the integral of A*f(A)dA from Amin to infinity is infinite when alpha LE 2. Now look at Figure 5(b). It shows that alpha is LE 2 quite often, and yet Figure 5(a) shows a finite mean floe area. It is not mathematically possible for a power-law distribution with the reported values of alpha to have a finite mean floe area. Fortunately, this is easily fixed by postulating that a maximum floe area (Amax) exists such that the distribution of floe areas follows a power law only for the finite range Amin LE A LE Amax. If this approach is taken, the authors should estimate Amax. If this approach is not taken, the authors should explain how to reconcile the mathematical contradiction.

Thank you for raising this point. We have been using a constant Amax value throughout our analysis. This value is now in the text and the choice of this value is now described in the manuscript (section 3.5).

Line 188. "The power law alpha is inversely correlated to the mean floe area". This finding is based on the data. It could also be checked analytically for a power-law distribution. As noted above, if alpha GT 2 then Amean = Amin * (alpha-1)/(alpha-2). As alpha becomes larger, Amean does indeed become smaller, although not linearly. But most of the values of alpha in this paper are LE 2. If the power law is defined on the finite interval [Amin,Amax] then Amean is a more complicated function of Amin, Amax, and alpha, but Amean still decreases as alpha increases, although not linearly. Whether the authors wish to include such an analytical comparison to the data is up to them, I am only pointing out the possibility to do so.

Thank you for detailing the mathematical functions behind this relationship. The authors agree that this derivation would complement the information presented in the manuscript. We have added an Appendix to derive the analytical mean and median which are now shown in Figure 6a and b.

Lines 210-218. This paragraph compares the power-law exponents of the present study to those found by Stern et al (2018a), but the comparison is not done correctly. The present study uses floe area as the measure of floe size, while Stern et al (2018a) used floe diameter. Assuming that floe area (A) is proportional to the square of floe diameter (x^2), a power-law distribution in A with exponent -alpha is equivalent to a power-law distribution in x with exponent -2*alpha+1. With this conversion, the authors should make a quantitative month-by-month comparison of their results with those of Perovich and Jones (2014), Hwang et al (2017), and Stern et al (2018a), all of which looked at the seasonal evolution of the FSD in the Beaufort Sea.

We agree. We have included a description of this conversion to the manuscript and added a quantitative comparison with the studies mentioned here (Section 5.2).

Regarding floe orientation (section 4.3), a couple of comments:

(1) It's possible that nearly circular floes could have a more variable orientation than elongated floes, because small changes in their perimeter could trigger large changes in their orientation. Therefore, as floe orientation becomes more variable (as in Figure 5(c) in June), how do we know whether it's due to floes becoming more randomly oriented or floes becoming more circular?

This is a good point. We looked at a subset of larger floes (>75th percentile), a subset of less circular floes (<25th percentile) and a subset of higher eccentricity (major axis/minor axis) floes (>75th percentile). We observed similar patterns in the evolution of the standard deviation of orientation to ensure that the pattern we were seeing is not just because floes are becoming smaller, more circular, and the eccentricity decreasing. We have added a sentence to the paper to address this comment.

Here is an example of the evolution of all floes (blue) and the subset of just higher eccentricity (orange) - the trend is less extreme, but still present.

[Figure]

all floes (blue), low circularity (green):

[Figure]

all floes (blues) large floes (red):

[Figure]

(2) Lines 227-231 discusses the orientation of large rectilinear floes, concluding that their orientation is less variable in the early season compared to the full data set, but increasingly variable into the summer. It's not clear what the point of this comparison is. The reader is left hanging, with no figures or numbers or explanation of the significance of the result.

Thank you for your feedback. Floe orientation is related to the generation of fractures on the ice as a result of the stress being applied (e.g., from atmospheric features, like the Beaufort High and anticyclonic winds) and the effects of coastal geometry. By looking at the standard deviation of floe orientation, we can infer at what point the ice movement transitions to free rotation- and if floe rotation can be linked to ocean vorticity underneath the ice. We have added text discussing floe orientation in more depth:

"The orientation of the floes is an indication of the stresses that have caused ice breakup and can provide insight into the structural properties of the ice pack to inform how it may respond to future stresses. When ice floes are able to rotate without interaction with other floes, their rotation rates can be related to ocean vorticity (Manucharyan et al., 2022)."

Minor Comments

First line of the Introduction. Consider changing "from the atmosphere to the ocean" to "between the atmosphere and the ocean" because the fluxes go both ways -- atm to ocean and ocean to atm.

Thank you, changed.

Lines 52-53. "The older and thicker multiyear ice is now melting out" -- but the loss of older and thicker multiyear ice has been going on for a long time. It was first noted in 2007:

Maslanik, J. A., C. Fowler, J. Stroeve, S. Drobot, J. Zwally, D. Yi, and W. Emery (2007), A younger, thinner Arctic ice cover: Increased potential for rapid, extensive sea-ice loss, Geophys. Res. Lett., 34, L24501, doi:10.1029/2007GL032043.

This phrase "now melting out" refers not to recent years, but to recent decades- as described in the study referenced at the end of this sentence that is specific to the Beaufort Sea. We removed this confusing phrase and added the Maslanik reference to the end of the sentence. Thank you for mentioning this study.

Lines 95-102. This same segmentation procedure was also used by Stern et al (2018a) -- see their section 3.2.

We added text to describe that this algorithm is based on Denton and Timmermans (2022) and Stern et al, (2018).

Lines 95-96. "where where"

Corrected

Lines 110-111. "from from" should be "range from"

Corrected

Line 119. "This quality assurance steps" -- singular or plural?

Singular, changed to "step"

Line 130. "The standard error of the fitted distribution" -- is this supposed to say the standard error of alpha?

Yes, thank you for catching that, corrected.

Line 150. "a correlation of 0.99" -- is this supposed to be a squared correlation? Figure 4 says "R^2 = 0.99"

Yes, corrected to "squared correlation"

Line 166. "The the"

Corrected.

Line 167. "values increases" -- singular or plural?

Changed to value increases

Figure 3 caption: Line 3. "imaeg" Line 4. Either write "captured" or delete the word "capture"

Corrected

Figure 5(b). The caption refers to "the slope anomaly" but the figure shows the actual slope, not the anomaly. Figure 5(c). The caption refers to "low circularity floes" in green but I don't see a green curve. Figure 5(d). The caption refers to "land (magenta)" but I don't see any magenta color. Figure 5(d). Is it really necessary to show all that gray area (clouds)? That makes it harder to see the green and blue areas, which are really the quantities of interest. Line 204. "Fig. 5b, shaded" -- I don't see any shaded part of Figure 5b.

Corrected. Cloud bars were removed, no shaded regions.

Lines 203-204. "the standard deviation calculated as in Equation 3." Is this supposed to say "standard error" or "standard deviation"? Does Equation 3 give the standard error or the standard deviation? See line 130.

Standard error. This has been revised.

Line 211. 59400 should be 594

Corrected.

Line 225. "that are can be"

Corrected.

Line 239. "increasing power law slopes" -- does this mean steeper or shallower? When alpha increases, the slope decreases (becomes more negative). When alpha decreases, the slope increases (becomes less negative). That's why it might be clearer to say steeper or shallower.

I see why this is confusing. We do not include the negative in the alpha- so increasing alpha means more negative and steepening slope. But this is not an increasing slope. Changed to increasing alpha (steepening power law slope)

Lines 241-242. This is the first mention of "discrete element models" in the paper. I would suggest that on line 37 the words "discrete element" be added just before the word "models".

Corrected.

Line 245. "new segmentation algorithm presented here" -- the segmentation algorithm is not new. See the comment above for lines 95-102.

Thank you for bringing this up. We changed this to "modified" and added a longer description in the introductory part of the methodology (Section 3) to describe how this has been adapted from previous work.

There are other minor typos not noted here.

The manuscript has been proofread to identify typos (see new version of the manuscript).

---

## Author Comment (AC2)

Reviewer 2:

The authors presented a study on sea ice floe size distribution in the Beaufort Sea based on the long-term MODIS dataset.
The method of ice floe segmentation is primarily based on the previously developed method by Paget et al. (2001). MODIS-derived ice floe segmentation was compared with the Sentinel-2 data based on three cases. They applied the method to 4,861 images of MODIS, identifying more than 9.4 million floes over 23 years. By analyzing the FSD in the Beaufort Sea over a long period, they unfolded the seasonal variability, including the decreasing mean floe area and increasing FSD power law slope as the summer progresses. Some major comments and specific comments are below for reference.

**Response:** We sincerely thank the reviewer for a thorough assessment of the manuscript. We believe that these comments and suggestions have improved the clarity of the presentation. We respond to the reviewer and describe changes to the manuscript in blue font below.

Major Comments

1.      Obviously, the title of the manuscript is too wide, and the authors only conducted analysis in the Beaufort Sea.

Thank you for bringing this up. We changed it to include "Beaufort Sea".

2.      The dataset is not well described. For instance, are all four bands of the S2 data used to segment ice floes?

Sentinel-2 has a total of 13 bands, four of which are at ten meter resolution. We apply the IFT to the Sentinel-2 imagery. The IFT uses the red channel for ice-water discrimination. We have added clarification to the description of the dataset.

3.      The methodology is too brief to be understood. Is the method exactly the same as that described by Paget et al. (2001)? Are any improvements achieved?

We now provide further details in the text on the erosion-expansion methodology in the introductory part of the methodology section. The methodology is not the same as Paget et al. (2001). Paget et al. (2001) performs an expansion-erosion algorithm. Stern et al. (2018) and Denton and Timmermans (2022) methods are also based on erosion-expansion operations but add an iterative component to identify different sizes of floes each round. Our algorithm builds on these two algorithms but differs in that the structuring element shape and the ice-water discrimination techniques were redesigned for MODIS imagery.

The text has been edited to highlight the importance of iteration in the expansion-erosion methodology:

"Paget et al. (2001) describes an erosion-expansion algorithm that erodes the boundaries of floes to separate them, subsequently regrowing them to their original shape. Denton and Timmermans (2022) and Stern et al. (2018a) introduce an iterative procedure that cycles through erosion-expansion varying the amount of erosion to identify floes of different sizes.

In this work, we build on the algorithm described in Denton and Timmermans (2022). We rewrite the algorithm in Python, and automate it to process thousands of images consecutively, introducing adaptive thresholds. The algorithm consists of four steps: pre-processing, ice-water discrimination, segmentation, and post-processing (Figure 2)."

Validation of the MODIS-derived sea ice floes is not convincing. On the one hand, only by presenting only three cases for nearly 5000 MODIS images seems to be inadequate. I would suggest that the authors add comparison experiments to provide a more robust argument.

Thank you for your comment. We need to keep in mind that the availability of validation imagery is very limited. Sentinel-2 is only available 20 km from the coast. Finding pairs of near-coincident cloud-free images in two different data sets (MODIS and Sentinel-2) is challenging. We present three different ice conditions to demonstrate the ability of the segmentation algorithm to operate in different SIC scenarios. Also each Sentinel-2 image is 110 km x 110 km, so the analysis of three Sentinel-2 images and coincident MODIS imagery allows for comparison of 36,300 km$^2$, a sufficient area for validation. We have added additional text to the manuscript to address uncertainties.

On the other hand, the current comparison itself is not convincing. As the three cases shown in Fig.3, there are obvious distinctions (e.g., the N values in the figure) between the two datasets. The MODIS data have a spatial resolution of 250 m, which is much larger than the S2 data of 10 m. Therefore, a threshold should be set to exclude those floes that MODIS cannot observe at all due to its coarse spatial resolution.

Thank you for bringing up this point. Given the different spatial resolutions, we set the minimum floe area as 5 km$^2$ for both the Sentinel-2 and the MODIS dataset. Even though more floes are identified in the Sentinel-2 imagery, the floes picked up in both data sets display similar FSDs. This highlights the value of MODIS to analyze FSDs dating back to the early 2000s.

Moreover, the validation shown in subsection 3.6 fails to "perform equally well". These ice floes, which are not segmented by MODIS, not only include the clustered loose ice but also some pack ice. Such a discrepancy is too visible to be explained by the different data resolution. The proposed method probably has limitations in terms of accuracy (however, it is hard to judge because the method is described too briefly). It is possible that some steps, such as the use of adaptive thresholds, excessive morphological processing, or the direct discarding of low-intensity floes, lead to unreasonable results.

We now provide additional information on the methodology as outlined above. We added a sentence to address that the algorithm performs well on both types of imagery, but does identify more floes in Sentinel-2 due to the higher resolution:
"Although the algorithm is able to identify smaller floes in the Sentinel-2 imagery, the floe sizes retrieved and the floe size distribution agree well between the Sentinel-2 and MODIS imagery."

We developed the algorithm based on the success of previous algorithms (Denton and Timmermans 2022, and Stern et al., 2018). The adaptive threshold is for accounting for variable but minimal low-level fog over the open water, as described in Section 3.2. Stern et al. (2018) also applies an adaptive threshold for ice-water discrimination. This adaptive

threshold is different but similar in purpose. Further, as noted in Section 3.5, since the low intensity floes account for less than 1% of floes, removing them does not introduce additional uncertainty.

4.      The title of section 4 should be narrowed down.

Thank you for your comment. We apologize if we are misunderstanding the comment, but we believe that the title of section 4 describes exactly what is included in the Section.  The title is "Spring to Summer Transition of Floe Characteristics," and we go on to describe how floe area, floe size distribution, and floe orientation evolve from March 1 to October 1.

I think that many relevant studies have been conducted with the FSD in the Arctic. Why not compare some previously derived FSD with the present results for further verification of its accuracy? Additionally, the statistical charts should be improved. For Fig. 5, it is fine to apply the "10-day running window", but one should also consider quantitative results such as scatters, boxplots, or upper and lower significance intervals to give the reader a clearer view of the author's raw statistical results. It would be better to try seasonal statistics, which might work better for the author's dataset (as too many MODIS data are excluded from analysis due to the cloud effect).

P2 L26 "see Stern et al. (2018b) for a comprehensive list of FSD studies": Even without a systematic review, a proper overview and summary of state of the art should be briefly presented here.

P2 L29-L31 "These studies have advanced our knowledge of seasonal evolution of the FSD ": The authors' review does not serve to summarise this knowledge.

Here we grouped together a few comments regarding the suggestion for more comparison to existing studies.

We discuss the spring to summer evolution seen in our data as well as in other studies. We have added more quantitative comparison with the Stern paper as well. (Section 5.2). We added a figure to show how the cloud fraction evolves over the summer. This figure also includes the number of floes identified in each 10 day running window to address the reviewers comment on more statistical information. The floe area mean, median and 75th percentile are shown in Figure 6. The standard deviation of orientation and alpha value are computed with all the floes in the 10-day running window and thus boxplots and scatters are not applicable. The standard error associated with the alpha value is discussed in Section 5.2.

Specific Comments

P2 L25 "floe size distribution": The full name has already been presented. Similar issues occurred several times in the manuscript, including "SIC", "SAR", and "MODIS". Please revise them.

Thank you for catching this detail. We have revised the text.

P2 L27-28 "... from radar imagery ... high-resolution imagery  ": The summary

of research types is weird. Isn't a SAR image a high-resolution image? Please rewrite this sentence to provide a detailed review of these studies.

We appreciate your attention to detail. We changed to: high-resolution optical satellite imagery.

P2 L35 "(Lopez Acosta et al., 2019) demonstrate..." -> "Lopez Acosta et al. (2019) demonstrate...". Please also note other similar citations.

Thank you, these have been revised.

P3 Fig.1 (c): Rather than showing the annual average ice floe numbers here, I'd be more interested in first finding out the annual use of MODIS data. In particular, long-term statistics need to know the amount of cloud-contaminated data for each year.
Figure 1c shows the total floe count per year. We also include a figure (Figure 5) and an introduction to Section 4 to discuss these statistics further.

P4 Fig.2: The authors presented a very clear case of pack ice. In addition, I would also like to see the algorithm's adaptability to high SIC, melting ice, MIZ, ice-water mixing, etc. After all, the authors aimed to focus on the phenomena related to the transition, which implies a rather complex ice condition.

We appreciate this suggestion, but kindly disagree. The purpose of Figure 2 is to illustrate the methodology. We have included more examples of the algorithm applied to a range of ice conditions in Figure 3.

P5 L90 "400-pixel (100 km) neighbourhood of a pixel subtracted by a constant": What is the constant?

Thank you for catching this typo. This has been revised for clarity. This was the description of the python function we used. But the default constant is zero.

The 400-pixel neighbourhood is a relatively large region. However, the masked MODIS contains many NAN values. How did the author choose the threshold at the edge of the NAN?

The threshold is applied to the masked image, so the NAN values are not included in the calculation for the threshold. We added a sentence: "At this point, the land and cloud pixels are masked and not considered in this step. "

Additionally, it would be better for the authors to argue for the rationality and specific benefits of adaptive thresholds where appropriate.

We agree with this comment. The justification for using adaptive thresholding is already described in the text: to consider variable pixel values of open water. We added a sentence to clarify this point: "In this way, we are able to account for varying brightness levels of open water given different ice concentrations and atmospheric conditions."

P5 3.3 section: Since erosion-expansion is an important step, it would be better to show the effect before and after morphological processing using the Fig.2 case.

And, what is the necessity of performing multiple iterations? Actually, in my opinion, besides the fact that it does improve the visual effect, too much morphological processing may lead to losing the original sea ice features.

This is a good point. We adapted this iterative erosion-expansion algorithm from Denton and Timmermans 2020 and Stern et al., 2018. Here, we review the original algorithms and focus on the changes we have made to apply similar methodologies to a different dataset at larger scales.

We interpret the second part of this question as why there are so many erosions within a round or why there are so many iterations. Hopefully we can clarify both.

In the first round, the erosion is very aggressive, so that small floes are eliminated, and the large floes are heavily eroded. Then the large floes that remain are tagged, regrown (dilated) the same number of times they were eroded, and then removed from the image. This iterative processes is repeated but with fewer erosions each round, so the size of the floes identified are decreasing in successive rounds.

In each round, the floes can be eroded multiple times since the erosion structuring element is small enough so that the features are not deformed. By removing just the border pixels we are maintaining the shape. And then this process is done multiple times to separate floes.

These methods are consistent with the iterative expansion-erosion algorithm presented in Denton and Timmermans, 2022 and Stern et al., 2018. Figure 3 in Stern et al., 2018 demonstrates the process:

[Figure]

**Figure Caption:**
MEDEA image processing steps. (a) Subset #1 of July 8, 2014 (within the red outline). See Figure 9 for location. This subset measures 3294 × 6537 pixels (3.3 × 6.5 km). (b) Binary image, first iteration. (c) Eroded and labeled image with holes filled, showing individual ice floes. (d) Re-grown ice floes in color, with unidentified ice floes in gray. (e) Binary image, second iteration, after the largest floes have been tagged and removed. (f) Final labeled image showing all identified floes. DOI: https://doi.org/10.1525/elementa.305.f3

We have text throughout Section 3 that references the methodologies in these studies and includes more statistics of the erosion-expansion algorithm.

P5 L103-L105 "On average, 26% of the classified sea ice area is identified as individual floes": Average of what?

Of all the images analyzed. This sentence is clarified: "Over the thousands of processed images, on average, 26% of the classified sea ice area is identified as individual floes, with remaining sections consisting of ice filaments, brash ice or pieces of ice smaller than the minimum detectable floe size."

P5 L111 "…the variation of the floe orientation": Which floes are the authors using circularity std to compare their variation? Are they ice floes that are tracked between time-series images, or are they all ice floes in the same image?

All ice floes in the 10-day running window of images. Added "within the 10-day window" to clarify

P6 L116-119: In fact, the low-intensity ice may be an important ice condition as well (especially in the transition, where it may represent a melting scenario). However, the author removed them outright, which would cause the subsequent results, especially for power law distributions, to be different from previous results. How can the authors justify this proposed step? Also, what are the units of 150? Can the authors prove <150 to be so-called brash ice rather than other types of sea ice (e.g. grey ice, melting ice)?

The pixel values are an 8-bit integer from 0-255. We look at the red channel as noted in the manuscript. 150 is determined empirically by testing various thresholds and examining the pixel brightness of floes, brash ice, and mixed pixels. Because of the size of each pixel (250 x 250 m), mixed pixels containing more than one surface type are present. The 150 intensity value ensures that low brightness ice floes are still included but mixed pixels over ambiguous surfaces are not included in this analysis. Also- as mentioned in the manuscript- this only removes <1% of the floes. The further analysis of how floe brightness intensity may represent the surface melt evolution is an interesting question but beyond the scope of this paper.

P6 L122-123 "that complements other parameters commonly used to describe ice floe fields, such as the sea ice concentration (SIC) and average ice thickness": I would suggest deleting this sentence.

We choose to retain the information as we think it is important to acknowledge there are multiple ways to describe the sea-ice cover, but have made edits to the paragraph in question. We have reworded this: "Taken together with the floe geometry properties outlined in Section 3.4, the FSD and other parameters commonly used to describe floe fields, such as the sea ice concentration (SIC) and average ice thickness, allow us to study the physical processes that shape the structure and evolution of sea ice."

P7 3.6 section: When using S2 images for segmentation, is there also a step to remove the <150 intensity? And, are the other parameter settings exactly the same (number of erosions and the calculation of adaptive thresholds)? What level of S2 data is used? These details may lead to a different adaptation of the ice floe segmentation algorithm to S2 data.

Thank you for bringing this up. We use Sentinel-2 Level 1C Top of Atmosphere Reflectance- this detail was added to the manuscript. The same algorithm was applied to both datasets, and there is a step to remove low intensity objects.

Fig 3&4: As I mentioned above, I don't think the comparison presents a good result. What is the "xmin"? Also, please present clearer quantitative results in the text.

We added the xmin value to Section 3.5. We have clarified the figure and added discussion on how the high sea ice concentration example exemplifies limitations of the algorithm. We hope that clarifications and additions elsewhere (including the addition of an Appendix deriving the analytical expressions) provide clearer quantitative results.

L149-151 "The areas of the matching 82 floes agree very well, with a correlation of 0.99,

and an absolute mean area difference of 0.18 km2 (Figure 4)": The authors only compared the 81 floes identified by both. I don't think this comparison is fair. The high-resolution S2 results should be treated as a reference, and all the S2-segmented floes should be compared to MODIS results. Obviously, the MODIS results have gross omissions.

The comparison of floe areas can only be done with the floes that are identified in both images. The power law distributions shown does include all floes identified in both the images that are in the MODIS range of detectable floe area. Here we are demonstrating that in the floe area range that MODIS can identify floes, there is strong agreement with the Sentinel-2 validation dataset. There is another question of the omission of smaller floes that I believe the reviewer may be referring to. We are carrying out a further error analysis to understand these limitations, but this is beyond the scope of this paper. Here we just assure that MODIS is acceptable for this range of floe sizes.

P7 L153-154 "...14% more floes are identified in the MODIS imagery compared to the Sentinel-2 imagery.": How can the author tell the 14%? I can only find that MODIS significantly underestimated the floe numbers.

We do not limit the floe size detected in the Sentinel-2 image. To compare likes, we sample the floes identified in the Sentinel-2 imagery that are in the MODIS size range. The 14% more floes are in the area range that MODIS can detect- so greater than 5 km$^2$. We have added a sentence to clarify that the Sentinel-2 identification includes floes less than 5 km$^2$. "as the identification of floes in the Sentinel-2 imagery was not limited to the MODIS range of floe sizes."

P7 L163 "We create 1000 bootstrap samples with replacement": I can't follow the author's point of this step.

This text has been modified for clarity. We include this step to show that the power law fit is a robust description of the dataset.

P7 L171 " SAR data is not as widely available": This sentence is misleading. If it refers specifically to long-term applications, SAR does have limitations. "other limitations or complications, such as speckle or granular noise". It is not correct. It is just the authors do not know how to deal with the valuable dataset.

Thank you for this point- we are referring to the availability of SAR data for this study which is a long term study that includes the summer time period. We have clarified the text: "However, SAR data is not as widely available for long-term applications, and has other limitations and complications, such as speckle, granular noise, and ambiguous returns when meltwater is present on the ice surface."

P8 L181-182: "…median floe size exhibits a similar pattern as the mean, with maximum median value of 17 km2 on 10 April...": The variations in the median are not visible at all, and this image should be modified.

Thank you for the comment- this image has been modified. A new panel has been added to separate the mean and the median.

P11 L210 "... Stern et al. (2018a) ...": Stern (2018a) included the analysis of small ice floes

(<5km2), right? So, it can be different from the authors' results.
In fact, in section 3.6, the alpha estimated from the MODIS is also lower than the S2 result.
It is suggested that the proposed algorithm is supposed to have some limitations. I would
suggest that the authors explain such reasons in detail.

Yes, Stern et al 2018 study includes floes as small as ~ 2.6 km$^2$. We have added a line:
"These values are similar and the small differences are likely due to location and time of the
observed floes." We acknowledge that this algorithm and dataset have limitations- this is
already described in Section 3.6 Validation and Limitations.

P12 L223 "…we find a low variability in their orientation (Fig. 5c)": Low variability? It
seems to me that Fig. 5(c) displays a clear increasing trend. Doesn't this suggest that as the
sea ice melts, the direction of the ice floes becomes more cluttered?

"this period" refers to the spring which was described in previous sentences. we replaced
"this period" with "the spring" for clarity.

P12 L223-224 "This effect is especially noticeable in areas of high ice concentration, where
the ice movement and readjustment to external forces is limited by the surrounding ice ":
It's hard to follow. If the authors mean that there is a decreasing tendency for std as SIC
increases, it should be expressed more clearly.
Otherwise, the most obvious phenomenon on this graph should be the very wide range of
std variation for SIC > 0.7.

Added for clarity "lower standard deviation of orientation with greater SIC"

P12 L227 "large rectilinear floes": This is less common usage, so please confirm.

Confirmed. Rectilinear means straight lines, and this is how we see the ice fracture in the
spring.

See also:

Arntsen, A. E., Song, A. J., Perovich, D. K., & Richter-Menge, J. A. (2015). Observations of
the summer breakup of an Arctic sea ice cover. *Geophysical Research Letters*, *42*(19), 8057-
8063.

where in the abstract they describe the fracturing in the ice pack resulting in rectilinear floes.

---

## Referee Report (RR1)

**Review on the manuscript "Seasonal evolution of the sea ice floe size distribution from two decades of MODIS data" by Buckly et al.**

Thank the authors for detailed responses to my previous review and they made a major revison to this manuscript. Most of the raised questions and comments were addressed. There are still some issues need to be further claified, mainly concerning the floe segmentation method and accuracy validation.

**A major comment:**

The improvement of this floe segmentation method compared to the previous methods is minimal. These fundamental algorithms were proposed based on a relative small amount of data or small coverages. Therefore, the present comparison and validation are still in doubt. Why the improved algorithm can be applied to a large amount of dataset and large spatial coverage are not very convinced. In their responses, it is mentioned that "Also each Sentinel-2 image is 110 km x 110 km, so the analysis of three Sentinel-2 images and coincident MODIS imagery allows for comparison of 36,300 km$^2$, a sufficient area for validation.", and "We need to keep in mind that the availability of validation imagery is very limited." However, these 36,300 km$^2$ is like "a drop in the ocean" compared with the three-years data in the vast Beaufort sea. I still suggest to add some quantitative methods to measure the effectiveness of the proposed segmentation method, rather than relying only on comparisons with the limited Sentinel-2 data.

**Specific comments:**

1.  P3 Fig.1 (c): Rather than showing the annual average ice floe numbers here, it might be more sutiable to show the data amount of processed MODIS in this figure. In particular, long-term statistics need to know the amount of cloud-contaminated data for each year.

2.  P6 L118: The authors mentioned in the revised article, "The morphological erosion operation is applied to the binary image, removing pixels on the object boundaries with a diamond-shaped structuring element with a radius of 1." Why is this

improvement more suitable for the task compared to previous methods? Why is the radius set to 1?

3. P7 section 3.6, the authors mentioned: "The same algorithm was applied to both datasets, and there is a step to remove low-intensity objects." For floes in the S2 data, is it reasonable to remove floes with an intensity lower than 150, just like in the MODIS data?

4. Despite the comparison with S2, as shown in Figure 3, the method has produced results with a significant amount of undetected floes. Consequently, conclusions drawn from the analysis based on these segmentation results would be greatly affected by the algorithm's performance. For the large amount of MODIS data that cannot be compared with S2 data, it is impossible to ensure the comprehensiveness of the detected floes. Therefore, the conclusions may be unreliable.

5. I am particularly interested in how floes are defined in areas with high sea ice concentration, as shown in Figures 3(g) and (h). How can we determine whether a floe ice is an independent entity rather than having re-grown together with other floes?

6. In the authors' response to the specific comments on L149-151 in the first-round review, they stated, "There is another question of the omission of smaller floes that I believe the reviewer may be referring to. We are carrying out a further error analysis to understand these limitations, but this is beyond the scope of this paper." In addition to the smaller floes, some large floes were also omitted.

---

## Author Response (AR2)

We sincerely thank the editor and the two anonymous reviewers for the helpful comments and suggestions. We addressed the raised comments in the manuscript, and include our responses to your questions in blue below. These revisions have enhanced the clarity of our paper and we thank the Editor and reviewers for bringing them to our attention.

**Editor questions and comments:**

Reviewer 1 emphasized the very narrow range of values for fitting a power law. I share these concerns and point to other methods to determine the fit, which represents further uncertainty (Muchow et al., 2021).

Muchow, M., Schmitt, A. U., and Kaleschke, L.: A lead-width distribution for Antarctic sea ice: a case study for the Weddell Sea with high-resolution Sentinel-2 images, The Cryosphere, 15, 4527–4537, https://doi.org/10.5194/tc-15-4527-2021, 2021.

Muchow et al. (2021) use two methods to determine the fit, the linear fit and the maximum likelihood fit as in Clauset et al., (2009).

For the linear fit, they reference Berk (2004), and state "that atypical values have a strong effect on the result." For this reason, we do not use a linear regression and stick to the maximum likelihood approach from Clauset et al. (2009). This method is widely used in many FSD studies as well as in the Muchow et al., (2021) paper. The powerlaw python package (Alstott et al., 2014) uses the statistics described by Clauset et al. (2009) and package the steps of fitting a power law in a reproducible, open methodology. This package has been used in many sea ice and iceberg size distribution studies for example (there are more beyond these):

Shiggins, C. J., Lea, J. M., & Brough, S. (2023). Automated ArcticDEM iceberg detection tool: insights into area and volume distributions, and their potential application to satellite imagery and modelling of glacier–iceberg–ocean systems. *The Cryosphere*, *17*(1), 15-32.

Walter, A., Lüthi, M. P., & Vieli, A. (2020). Calving event size measurements and statistics of Eqip Sermia, Greenland, from terrestrial radar interferometry. *The Cryosphere*, *14*(3), 1051-1066.

Köhler, A., Pętlicki, M., Lefeuvre, P. M., Buscaino, G., Nuth, C., & Weidle, C. (2019). Contribution of calving to frontal ablation quantified from seismic and hydroacoustic observations calibrated with lidar volume measurements. *The Cryosphere*, *13*(11), 3117-3137.

Denton, A. A., & Timmermans, M. L. (2022). Characterizing the sea-ice floe size distribution in the Canada Basin from high-resolution optical satellite imagery. *The Cryosphere*, *16*(5), 1563-1578.

In our work we follow the full methodology of the powerlaw package which includes testing to determine if the data do indeed fit a power law distribution with the log-likelihood ratio of various distribution's fit to the data. We test: power law, exponential, and lognormal distributions.

I find the description of the illustration of the segmentation results not sufficient. Especially the fact that large ice floes are sometimes not separated by the segmentation and form larger objects of the same color. What does this mean for the resulting FSD?

We have added discussion of algorithm limitations to a new Section 3.7 Limitations.

Where does the number of 9.4 million floes come from? Are any of them counted multiple times?

The 9.4 million floes are not unique identifications. This number reports all identified floes, treating each image as an independent observation. We added the clause: "\ref{fig:loc}b; note that these are not 9.4 unique floes as each image is taken as an independent observation."

How is the ice floe tracker IFT used in this study? It was mentioned in the introduction and I expected that it is used for some purpose, e.g. to avoid double counting. Please clarify.

This work was motivated by expanding on the recently developed IFT, which provides unique sea ice floe observations from the MODIS dataset. We agree this is not clearly explained and not essential to include as currently worded. We have reworded the introduction to describe the IFT as a previous study and in the conclusions we suggest incorporating the new segmentation algorithm into the tracking of the IFT pipeline.

What information is stored in the "floe library"? Please describe this library in more detail. Is this a data set? See https://www.the-cryosphere.net/policies/data\_policy.html

The floe library dataset has been uploaded and documented on Zenodo- we added this to the data availability section.

Section 3.5: 97% of the floes fall in this size range. Where does this number come from? Does this percentage mean by area coverage? Or by the number of floes? Is this the same for MODIS and S2?

By number of floes. We added a clarification here. This is referring to the MODIS floes. We provide statistics for the Sentinel-2 floes when Sentinel-2 is introduced in the next section.

Section 5: the imagery largest record of Earth to date. Really? What about Landsat? Maybe a question of definition?

Landsat has acquired imagery since 1972, but primarily of Earth's land surface. The coverage allows for an 8-day repeat cycle of each Landsat scene. MODIS, on the other hand, is the longest continuous daily global satellite observation record ever compiled. We added clarification in the text.

Most references are incomplete, the DOIs are missing. This would result in a long list of complaints for copy editing.

Thank you for pointing this out, we have added the DOIs for all references.

Figure 3cfi: Better plot data points (with error bars) instead connected lines

We have now plotted points for each floe area bin. We have not plotted error bars as it is not obvious what statistic would be optimal here for the binned data. One approach may be to plot the bootstrapped samples, but we hope for the sake of clarity that the uncertainty on the power law fit alpha will suffice.

You can upload your code+data already now on Zenodo with restricted access.

Added code in the data availability section.

**Reviewer #1.**

The authors have addressed all of my comments. I have a few further comments about the revised manuscript.

Line 87. "on average 58% of an image is covered in opaque clouds..." -- The first draft of this paper said 71%. Why did it change to 58%?

After submission, we noticed the original calculation included the land mask, giving the fraction of the area masked by land or cloud, not just cloud. The 58% is the percentage of the image covered in clouds.

Line 142. The floe areas considered in this work range from 5 km2 to 300 km2. That's only a factor of 60, and the range of floe diameters is only sqrt(60) < 8. This is a VERY narrow range of values for fitting a power law -- less than one order of magnitude -- and is necessarily accompanied by a high degree of uncertainty as to whether a power law is indeed the best-fitting function. The authors should acknowledge that the range of floe sizes is very narrow for fitting a power law.

We refer the reviewer to our response to the editor on this point (above). We follow the full methodology of the powerlaw package which includes testing to determine if the data fit a power law distribution with the log-likelihood ratio of various distribution's fit to the data. We test: power law, exponential, and lognormal distributions and determine that the power law distribution is indeed the best fit.

We are limited by the minimum and maximum floe sizes we can accurately detect. The smallest floe size is determined by sensor resolution. To be consistent in fitting the power law throughout the season, we use 300 km2 as the maximum floe size. The maximum floe size is somewhat constrained by the image size. The larger a floe is, the more likely it is to intersect with the image border, and our algorithm eliminates floes that intersect with the border because the floe area statistics are then inaccurate. Thus, the number of large floes is underestimated.

We provide the low values of the sigma values associated with the power law fit. We have made clear in the paper the range of floe sizes to which this power law fit is applied. A number of other studies include

analysis of a small range of floe sizes with the range of floe max:min floe diameters e.g: (Holt and Martin (2001) ratio 11.1, Inuoe et al., (2004) ratio 7.6, Hwang et al., (2017) ratio 6.7).

Data limitations are such that our power law fits are necessarily over a narrow range of values, 5 to 300 km2. It may be that the fits presented here do not fully represent the sea ice in areas of high ice concentration, but we show a robust seasonal trend in floe size distribution in the floes size 5 to 300 km2. We have added a discussion of the limitations of the floe size range analyzed in Section 3.7 Data and Algorithm Limitations.

It may be useful to see how the fit differs for different ranges of floe sizes. In the figure below we show the FSD fit to floes with area  $5 \le x \le 300 \text{ km}^2$  (left panel) and floes  $x \ge 5 \text{ km}^2$  - i.e. all floes because floes with  $x \le 5 \text{ km}^2$  are not identified (right panel). The maximum floe size is 321, 813 km2. 97% of identified floes are in the fitted range ( $5 \le x \le 300 \text{ km}^2$ ). Although the alpha values are similar, in the right panel we see the true FSD (solid line) deviates from the power law fit (dotted) around floes of size  $300 \text{ km}^2$  (vertical line).

Equation 3 and following -- is alpha required to be greater than 1? It seems that equation 3 only requires alpha not equal to 1, but equation 4 does apparently require alpha greater than 1, otherwise the uncertainty would be negative.

**We added a note that alpha must be greater than 1.**

Line 173. "The difference in alpha values ranges from 0.03 to 0.21" -- When I look at Figure 3(c,f,i) I see differences of 0.06 (3c), 0.25 (3f), and 0.17 (3i). Where does "0.03 to 0.21" come from?

Corrected to 0.06 to 0.25. Thank you for noticing that.

Line 191. "bootstrap with replacement" should be "sampling with replacement"

**Corrected**

Lines 183-194. About the bootstrapping procedure, my understanding is as follows. Start with a randomly selected image that contains N floes. From that collection of floes, select N floes WITH replacement. If the sampling had been WITHOUT replacement, then there would be no difference between the original set of floes and the bootstrapped set. The only randomization in this bootstrapping procedure is the "with replacement" sampling scheme. That hardly seems like a proper bootstrap to me.

Our sampling has been done with replacement as described. This is a powerful approach in data analysis that gives us more confidence in the statistics for a number of reasons. Our large number of bootstrap samples (1000 for each image) gives us additional confidence in alpha.

Furthermore, if the intention is to assess the uncertainty in the exponent alpha, that should come from the fitting procedure itself (most packages include an uncertainty estimate for the estimated parameters) or from a goodness-of-fit test. Why is bootstrapping even needed?

We include the bootstrapping as an additional method of estimating uncertainty. We have also included the uncertainty in the exponent alpha that is provided by the powerlaw package.

Lines 231-232. "The mean and median OF THE FITTED POWER LAW can be expressed analytically." Corrected.

Line 239. "Fig 6b" should be "Fig 6c" Corrected.

Line 254. "50 m to 5 km2" -- Is this supposed to say 50 m2? Are the units supposed to be diameter or area? Ver  $50 \text{ m}^2$  this has been corrected.

Yes,  $50 \text{ m}^2$ , this has been corrected.

Line 268. "Fig 6c, shaded" -- I don't see shading in Fig 6c. Corrected, removed "shaded"

Appendix A. In A1, does alpha have to be greater than 1? In A2, I think it would be useful to plug in the expression for c from A1 and write E as  $((1-a)/(2-a))^*((xmax^{(2-a)-xmin^{(2-a)}})/(xmax^{(1-a)-xmin^{(1-a)}}))$  where a=alpha.

Yes, alpha must be greater than 1, now noted. We now provide this expression (Equation A6).

Note than alpha cannot equal 1 or 2. Those are special cases. We have similarly added after (A6): "for \alpha \ne 2.".

In A3, I think it would be useful to plug in the expression for c from A1 and write x0 as  $((1/2)*(xmax^{(1-a)+xmin^{(1-a)}}))^{(1/(1-a))}$ Again alpha=1 is a special case. Thank you, we have added this (Equation A9).

**Reviewer 2:**

Review on the manuscript "Seasonal evolution of the sea ice floe size distribution from two decades of MODIS data" by Buckly et al.

Thank the authors for detailed responses to my previous review and they made a major revison to this manuscript. Most of the raised questions and comments were addressed. There are still some issues need to be further claified, mainly concerning the floe segmentation method and accuracy validation.

**Thank you for your helpful comments, we have added a Section 3.7 to further detail the limitations of the algorithm.**

A major comment: The improvement of this floe segmentation method compared to the previous methods is minimal. These fundamental algorithms were proposed based on a relative small amount of data or small coverages. Therefore, the present comparison and validation are still in doubt. Why the improved algorithm can be applied to a large amount of dataset and large spatial coverage are not very convinced. In their responses, it is mentioned that "Also each Sentinel-2 image is 110 km x 110 km, so the analysis of three Sentinel-2 images and coincident MODIS imagery allows for comparison of 36,300 km2, a sufficient area for validation.", and "We need to keep in mind that the availability of validation imagery is very limited." However, these 36,300 km2 is like "a drop in the ocean" compared with the three-years data in the vast Beaufort sea. I still suggest to add some quantitative methods to measure the effectiveness of the proposed segmentation method, rather than relying only on comparisons with the limited Sentinel-2 data.

We recognize there are limitations to this dataset and methodology, such as the splitting of floes in high ice concentration areas, and the inability to detect the smallest floes due to sensor resolution. We hope this has been sufficiently addressed in our added section. It seems somewhat arbitrary that the validation imagery must cover a certain percentage of the total area analyzed. Rather, we ensured an appropriate validation via the use of a selection of varied sea-ice states (i.e., a range of sea-ice conditions in the Sentinel-2 imagery).

**Specific comments:**

1. P3 Fig.1 (c): Rather than showing the annual average ice floe numbers here, it might be more sutiable to show the data amount of processed MODIS in this figure. In particular, long-term statistics need to know the amount of cloud-contaminated data for each year.

**Please see Figure 5 for those statistics.**

2. P6 L118: The authors mentioned in the revised article, "The morphological erosion operation is applied to the binary image, removing pixels on the object boundaries with a diamond-shaped structuring element with a radius of 1." Why is this improvement more suitable for the task compared to previous methods? Why is the radius set to 1?

This is the same structuring element used by Denton and Timmermans, 2022; we note this as part of our description of the methodology, not to suggest that this is necessarily an improvement on previous methods.

3. P7 section 3.6, the authors mentioned: "The same algorithm was applied to both datasets, and there is a step to remove low-intensity objects." For floes in the S2 data, is it reasonable to remove floes with an intensity lower than 150, just like in the MODIS data?

**We use identical algorithms for both data types for appropriate comparison of the results.**

4. Despite the comparison with S2, as shown in Figure 3, the method has produced results with a significant amount of undetected floes. Consequently, conclusions drawn from the analysis based on these segmentation results would be greatly affected by the algorithm's performance. For the large amount of MODIS data that cannot be compared with S2 data, it is impossible to ensure the comprehensiveness of the detected floes. Therefore, the conclusions may be unreliable.

We believe we have sufficiently validated the algorithm while acknowledging its deficiencies. Please see the new Section 3.7 Limitations for an in-depth discussion of the limitations of this algorithm.

5. I am particularly interested in how floes are defined in areas with high sea ice concentration, as shown in Figures 3(g) and (h). How can we determine whether a floe ice is an independent entity rather than having re-grown together with other floes?

This is not only a question about how the algorithm handles this but physically how we define an individual floe, which is subjective to a certain extent. The refreezing of leads and fusion of floes into a single floe is complicated. For example, at what point in the fusion process should we consider multiple floes as a single floe? Further, this question is probably answered differently depending on the end use of the floe size distribution. For example, considering how a floe field reacts to an ocean swell, a weak fusion might as well be two independent floes, because it will break easily. When considering the distribution of upper-ocean solar heating, adjacent floes that are not necessarily fused but do not allow the penetration of light into the ocean, in this case may appropriately be considered as a single floe. This dilemma also bolsters the case for considering floe size distribution in context of other defining ice characteristics such as ice concentration and lead fraction. Any floe segmentation algorithm will be limited by the resolution of imagery and the sensor sensitivity to leads.

We split Section 3.6 Validation and Limitations into 3.6 Validation and 3.7 Limitations. In Section 3.7, we discuss limitations of the algorithm such as:

- Delineation of large floes in high sea ice concentration areas
- The undersampling of large floes due to image size and separation
- Limitations on the power law fit related to the limited range of sea ice floe areas

6. In the authors' response to the specific comments on L149-151 in the first-round review, they stated, "There is another question of the omission of smaller floes that I believe the reviewer may be referring to.

We are carrying out a further error analysis to understand these limitations, but this is beyond the scope of this paper." In addition to the smaller floes, some large floes were also omitted.

Thanks for the comment. We have included discussion in the new Section 3.7 on the large floes that are omitted.